# A new perspective on building efficient and expressive 3D equivariant graph neural networks

**Weitao Du**[*]
Chinese Academy of Sciences
duweitao@mass.ac.cn

**Yuanqi Du**[*]
Cornell University
yd392@cornell.edu

**Limei Wang**[*]
Texas A&M University
limei@tamu.edu

**Dieqiao Feng**
Cornell University

**Guifeng Wang**
Zhejiang University

**Shuiwang Ji**
Texas A&M University

**Carla P Gomes**
Cornell University

**Zhi-Ming Ma**
Chinese Academy of Sciences

## Abstract

Geometric deep learning enables the encoding of physical symmetries in modeling 3D objects. Despite rapid progress in encoding 3D symmetries into Graph Neural Networks (GNNs), a comprehensive evaluation of the expressiveness of these network architectures through a local-to-global analysis lacks today. In this paper, we propose a local hierarchy of 3D isomorphism to evaluate the expressive power of equivariant GNNs and investigate the process of representing global geometric information from local patches. Our work leads to two crucial modules for designing expressive and efficient geometric GNNs; namely local substructure encoding (**LSE**) and frame transition encoding (**FTE**). To demonstrate the applicability of our theory, we propose LEFTNet which effectively implements these modules and achieves state-of-the-art performance on both scalar-valued and vector-valued molecular property prediction tasks. We further point out future design space for 3D equivariant graph neural networks. Our codes are available at https://github.com/yuanqidu/LeftNet.

## 1   Introduction

The success of many deep neural networks can be attributed to their ability to respect physical symmetry, such as Convolutional Neural Networks (CNNs) [1] and Graph Neural Networks (GNNs) [2]. Specifically, CNNs encode translation equivariance, which is essential for tasks such as object detection. Similarly, GNNs encode permutation equivariance, which ensures that the node ordering does not affect the output node representations, by aggregating neighboring messages. Modeling 3D objects, such as point clouds and molecules, is a fundamental problem with numerous applications [3], including robotics [4], molecular simulation [5, 6], and drug discovery [7–11]. Different from 2D pictures and graphs that only possess the translation [1] and permutation [2] symmetry, 3D objects intrinsically encode the complex $SE(3)/E(3)$ symmetry [12], which makes their modeling a nontrivial task in the machine learning community.

To tackle this challenge, several approaches have been proposed to effectively encode 3D rotation and translation equivariance in the deep neural network architectures, such as TFN [13], EGNN [14], and SphereNet [15]. TFN leverages spherical harmonics to represent and update tensors equivariantly,

---

[*]Equal contribution.

37th Conference on Neural Information Processing Systems (NeurIPS 2023).

while EGNN processes geometric information through vector update. On the other hand, SphereNet is invariant by encoding scalars like distances and angles. Despite rapid progress has been made on the empirical side, it's still unclear what 3D geometric information can equivariant graph neural networks capture and how the geometric information is integrated during the message passing process [16–18]. This type of analysis is crucial in designing expressive and efficient 3D GNNs, as it's usually a trade-off between encoding enough geometric information and preserving relatively low computation complexity. Put aside the $SE(3)/E(3)$ symmetry, this problem is also crucial in analysing ordinary GNNs. For example, 1-hop based message passing graph neural networks [19] are computationally efficient while suffering from expressiveness bottlenecks (comparing with subgraph GNNs [20, 21]). On the other hand, finding a better trade-off for 3D GNNs is more challenging, since we must ensure that the message updating and aggregating process respects the $SE(3)/E(3)$ symmetry.

In this paper, we attempt to discover better trade-offs between computational efficiency and expressive power for 3D GNNs by studying two specific questions: 1. What is the geometric expressiveness of 3D GNNs through a local 3D graph isomorphism lens? 2. What is expressiveness of 3D GNNs in representing global geometric information through local patches? The first question relates to the design of node-wise geometric messages, and the second question relates to the design of equivariant (or invariant) aggregation. To tackle these two problems, we take a local-to-global approach. More precisely, we first define three types of 3D graph isomorphism to characterize local 3D structures: tree, triangular, and subgraph isomorphism, following a local hierarchy. Our local hierarchy lies between the 1-hop and 2-hop geometric isomorphism defined in Joshi et al. [22], detailed in Appendix G.2; thus, it can be used to measure the expressive power of 3D GNNs by their ability of differentiating non-isomorphic 3D structures similar to the geometric WL test proposed in Joshi et al. [22]. Under this theoretical framework, we summarize one essential ingredient for building expressive geometric messages on each node: local 3D substructure encoding (**LSE**), which allows an invariant realization. To answer the second question, we analyze whether local invariant features are sufficient for expressing global geometries by message aggregation, and it turns out that frame transition encoding (**FTE**) is crucial during the local to global process. Although **FTE** can be realized by invariant scalars, we further demonstrate that introducing equivariant messaging passing is more efficient. After presenting **LSE** and **FTE** modules, we are able to present a modular overview of 3D GNNs designs. In realization of our theoretical findings, we propose LEFTNet that efficiently implements **LSE** and **FTE** without sacrificing expressiveness. Empirical experiments on real-world scenarios, predicting scalar-valued property (e.g. energy) and vector-valued property (e.g. force) for molecules, demonstrate the effectiveness of LEFTNet.

## 2 Preliminary

In this section, we provide an overview of the mathematical foundations of $E(3)$ and $SE(3)$ symmetry, which is essential in modeling 3D data. We also summarize the message passing graph neural network framework, which enables the realization of $E(3)/SE(3)$ equivariant models.

**Euclidean Symmetry.** Our target is to incorporate Euclidean symmetry to ordinary permutation-invariant graph neural networks. The formal way of describing Euclidean symmetry is the group $E(3) = O(3) \rtimes T(3)$, where $O(3)$ corresponds to reflections (parity transformations) and rotations. For tasks that are anti-symmetric under reflections (e.g. chirality), we consider the subgroup $SE(3) = SO(3) \rtimes T(3)$, where $SO(3)$ is the group of rotations. We will use $SE(3)$ in the rest of the paper for brevity except when it's necessary to emphasize reflections.

**Equivariance.** A tensor-valued function $\varphi(\mathbf{x})$ is said to be **equivariant** with respect to $SE(3)$ if for any translation or rotation $g \in SE(3)$ acting on $\mathbf{x} \in \mathbf{R}^3$, we have

$$\varphi(g\mathbf{x}) = \mathcal{M}(g)\varphi(\mathbf{x}),$$

where $\mathcal{M}(\cdot)$ is a matrix representation of $SE(3)$ acting on tensors. See Appendix A for a general definition of tensor fields. In this paper, we will use **bold** letters to represent an equivariant tensor, e.g., $\mathbf{x}$ as a position vector. It is worth noting that when $\varphi(\mathbf{x}) \in \mathbf{R}^1$ and $\mathcal{M}(g) \equiv 1$ (the constant group representation), the equivariant function $\varphi(\mathbf{x})$ is also called an **invariant** scalar function.

**Message Passing Scheme for Geometric Graphs.** A geometric graph $G$ is represented by $G = (V, E)$. Here, $v_i \in V$ denotes the set of nodes (vertices, atoms), and $e_{ij} \in E$ denotes the set of edges. For brevity, the edge feature attached on $e_{ij}$ is also denoted by $e_{ij}$. $\mathbf{X} = (\mathbf{x}_1, \ldots, \mathbf{x}_n) \in \mathbf{R}^{n \times 3}$ denotes the node positions which determine the geometric structure of $G$.

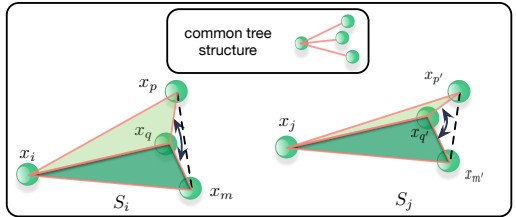 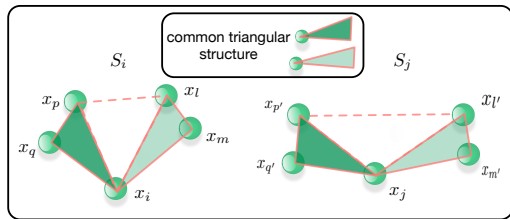

(a)  (b)

Figure 1: (a) $\mathbf{S}_i$ and $\mathbf{S}_j$ share the same tree structure (edge lengths are identical), but they are not triangular isomorphic (different dihedral angles); (b) $\mathbf{S}_i$ and $\mathbf{S}_j$ are triangular isomorphic but not subgraph isomorphic (the relative distance between the two triangles is different).

A common machine learning tool for modeling graph-structured data is the Message Passing Neural Network (MPNN) [16]. A typical 1-hop MPNN framework consists of two phases: (1) message passing; (2) readout. Let $h_i^l, h_j^l$ denote the $l$-th layer's node features of source $i$ and target $j$ that also depend on the 3D positions $(\mathbf{x}_i, \mathbf{x}_j)$, then the aggregated message is

$$m_i^l = \bigoplus_{j \in \mathcal{N}(i)} m_{ij}(h^l(\mathbf{x}_i), h^l(\mathbf{x}_j), e_{ij}^l), \tag{1}$$

and $\bigoplus_{j \in \mathcal{N}(i)}$ is any permutation-invariant pooling operation between the 1-hop neighbors of $i$. We also include the edge features $e_{ij}^l$ into the message passing phase for completeness. 3D **equivariant** GNNs (3D GNNs for short) require the message $m_i$ to be equivariant with respect to the geometric graph. That is, for an arbitrary edge $e_{ij}$:

$$m_{ij}(h^l(g\mathbf{x}_i), h^l(g\mathbf{x}_j)) = \mathcal{M}(g)m_{ij}(h^l(\mathbf{x}_i), h^l(\mathbf{x}_j)), \tag{2}$$

where $g \in SE(3)$ is acting on the whole geometric graph simultaneously: $(\mathbf{x}_1, \ldots, \mathbf{x}_n) \rightarrow (g\mathbf{x}_1, \ldots, g\mathbf{x}_n)$. For example, the invariant model ComENet [23] satisfies Eq. 2 by setting $\mathcal{M}(g) \equiv 1$, and MACE [24] realized Eq. 2 for nonconstant irreducible group representations $\mathcal{M}(g)$ through spherical harmonics and Clebsch-Gordan coefficients.

## 3 A local hierarchy of 3D graph isomorphism

As presented in Section 2, defining expressive messages is an essential component for building powerful 3D GNNs. In this section, we develop a fine-grained characterization of local 3D structures and build its connection with the expressiveness of 3D GNNs.

Since the celebrated work [25], a popular expressiveness test for permutation invariant graph neural networks is the 1-WL graph isomorphism test [26], and Wijesinghe and Wang [27] has shown that the 1-WL test is equivalent to the ability to discriminate the **local** subtree-isomorphism. It motivates us to develop a novel (local) 3D isomorphism for testing the expressive power of 3D GNNs. However, this task is nontrivial, since most of the previous settings for graph isomorphism are only applicable to 2D topological features. For 3D geometric shapes, we should take the $SE(3)$ symmetry into account. Formally, two 3D geometric graphs $\mathbf{X}, \mathbf{Y}$ are defined to be **globally** isomorphic, if there exists $g \in SE(3)$ such that

$$\mathbf{Y} = g\mathbf{X}. \tag{3}$$

In other words, $\mathbf{X}$ and $\mathbf{Y}$ are essentially the same, if they can be transformed into each other through a series of rotations and translations. Not that Eq. 3 is up to the permutation of nodes. Inspired by Wijesinghe and Wang [27], now we introduce a novel hierarchy of $SE(3)$ equivariant local isomorphism to measure the local similarity of 3D structures.

Let $\mathbf{S}_i$ represent the 3D subgraph associated with node $i$. This subgraph contains all the 1-hop neighbors of $i$ as its node set, along with all edges in $E$ where both end points are one-hop neighbors of $i$. For each edge $e_{ij} \in E$, the mutual 3D substructure $\mathbf{S}_{i-j}$ is defined by the intersection of $\mathbf{S}_i$ and $\mathbf{S}_j$: $\mathbf{S}_{i-j} = \mathbf{S}_i \cap \mathbf{S}_j$.

Given two local subgraphs $\mathbf{S}_i$ and $\mathbf{S}_j$ that correspond to two nodes $i$ and $j$ (not necessarily adjacent), we say $\mathbf{S}_i$ is {tree-, triangular-, subgraph-} isometric to $\mathbf{S}_j$, if there exists a bijective function

$f : \mathbf{S}_i \to \mathbf{S}_j$ such that $h_{f(u)} = h_u$ for every node $u \in \mathbf{S}_i$, and the following conditions hold respectively:

- **Tree Isometric:** If there exists a collection of group elements $g_{iu} \in SE(3)$, such that $(\mathbf{x}_{f(u)}, \mathbf{x}_{f(i)}) = (g_{iu}\mathbf{x}_u, g_{iu}\mathbf{x}_i)$ for each edge $e_{iu} \in \mathbf{S}_i$;
- **Triangular Isometric:** If there exists a collection of group elements $g_{iu} \in SE(3)$, such that the corresponding mutual 3D substructures satisfy: $\mathbf{S}_{f(u)-f(i)} = g_{iu}\mathbf{S}_{u-i}$ for each edge $e_{iu} \in \mathbf{S}_i$;
- **Subgraph Isometric:** for any two adjacent nodes $u, v \in \mathbf{S}_i$, $f(u)$ and $f(v)$ are also adjacent in $\mathbf{S}_j$, and there exist a single group element $g_i \in SE(3)$ such that $g_i\mathbf{S}_i = \mathbf{S}_j$.

Note that tree isomorphism only considers edges around a central node, which is of a tree shape. On the other hand, the mutual 3D substructure can be decomposed into a bunch of triangles (since it's contained in adjacent node triplets), which explains the name of triangular isomorphism.

In fact, the three isomorphisms form a hierarchy from micro to macro, in the sense that the following implication relation holds:



**Subgraph Isometric $\Rightarrow$ Triangular Isometric $\Rightarrow$ Tree Isometric**



This is an obvious fact from the above definitions. To deduce the reverse implication relation, we provide a visualized example. Fig. 1 shows two examples of local 3D structures: 1. the first one shares the same tree structure, but is not triangular-isomorphic; 2. the second one is triangular-isomorphic but not subgraph-isomorphic. In conclusion, the following diagram holds:



**Tree Isometric $\nRightarrow$ Triangular Isometric $\nRightarrow$ Subgraph Isometric**



One way to formally connect the expressiveness power of a geometric GNN with their ability of differentiating geometric subgraphs is to define geometric WL tests, the reader can consult [22]. In this paper, we take an intuitive approach based on our nested 3D hierarchy. That is, if two 3D GNN algorithms A and B can differentiate all non-isomorphic local 3D shapes of tree (triangular) level, while A can differentiate at least two more 3D geometries which are non-isomorphic at triangular(subgraph) level than B, then we claim that algorithm A's expressiveness power is more powerful than B.

Since tree isomorphism is determined by the one-hop Euclidean distance between neighbors, distinguishing local tree structures is relatively simple for ordinary 3D equivariant GNNs. For example, the standard baseline SchNet [28] is one instance of Eq. 1 by setting $e_{ij}^t = \mathbf{RBF}(d(\mathbf{x}_i, \mathbf{x}_j))$, where $\mathbf{RBF}(\cdot)$ is a set of radial basis functions. Although it is powerful enough for testing tree non-isomorphism (assuming that $\mathbf{RBF}(\cdot)$ is injective), we prove in Appendix D that SchNet cannot distinguish non-isomorphic structures at the triangular level.

On the other hand, Wijesinghe and Wang [27] has shown that by leveraging the topological information extracted from local overlapping subgraphs, we can enhance the expressive power of GNNs to go beyond 2D subtree isomorphism. In our setting, the natural analogue of the overlapping subgraphs is exactly the mutual 3D substructures. Now we demonstrate how to merge the information from 3D substructures to the message passing framework (1). Given an $SE(3)$-**invariant** encoder $\phi$, define the 3D structure weights $A_{ij} := \phi(\mathbf{S}_{i-j})$ for each edge $e_{ij} \in E$. Then, the message passing framework (1) is generalized to:

$$m_i^l = \bigoplus_{j \in \mathcal{N}(i)} m_{ij}(h^l(\mathbf{x}_i), h^l(\mathbf{x}_j), A_{ij}h^l(\mathbf{x}_j), e_{ij}^l). \tag{4}$$

Formula 4 is an efficient realization of enhancing 3D GNNs by injecting the mutual 3D substructures. However, a crucial question remains to be answered: *Can the generalized message passing framework boost the expressive power of 3D GNNs?* Under certain conditions, the following theorem provides an affirmative answer:

**Theorem 3.1.** *Suppose $\phi$ is a universal SE(3)-invariant approximator of functions with respect to the mutual 3d structures $\mathbf{S}_{i-j}$. Then, the collection of weights $\{\{A_{ij} := \phi(\mathbf{S}_{i-j})\}_{e_{ij} \in E}\}$ enables the differentiation of local structures beyond tree isomorphism. Furthermore, under additional injectivity assumptions (as described in Eq. 16), 3D GNNs based on the enhanced message passing framework (see Section 4) map at least two distinct local 3D subgraphs with isometric local tree structures to different embeddings.*

*Proof.* The detailed proof is provided in Appendix D. □

This theorem confirms that the enhanced 3D GNN (formula 4) is more expressive than the SchNet baseline, at least in testing local non-isomorphic geometric graphs. The existence of such local encoder $\phi$ is proved in two ways : 1. Equivariant construction by the Atomic Cluster Expansion (ACE) [29]; 2. Invariant construction under scalarization by edge-wise equivariant frames [30]. Note that there are other different perspectives on characterizing 3D structures, we will also briefly discuss them in Appendix D.

## 4 From local to global: the missing pieces

In the last section, we introduced a local 3D graph isomorphism hierarchy for testing the expressive power of 3D GNNs. Furthermore, we motivated adding a SE(3)-**invariant** encoder to improve the expressive power of one-hop 3D GNNs by scalarizing not only pairwise distances but also their mutual 3D structures in Theorem 3.1. However, to build a powerful 3D GNN, it remains to be analyzed how a 3D GNN acquires higher order (beyond 1-hop neighbors) information by accumulating local messages. A natural question arises: *Are local invariant messages enough for representing **global** geometric information?*

To formally formulate this problem, we consider a two-hop aggregation case. From fig. 2, the central atom $a$ is connected with atoms $b$ and $c$. Except for the common neighbor $a$, other atoms that connect to $b$ and $c$ form two 3D clusters, denoted by **B**, **C**. Suppose the ground-truth interaction potential of **B** and **C** imposed on atom $a$ is described by a tensor-valued function $f_a(\mathbf{B}, \mathbf{C})$. Since **B** and **C** are both beyond the 1-hop neighborhood of $a$, the information of $f_a(\mathbf{B}, \mathbf{C})$ can only be acquired after two steps of message passing: 1. atoms $b$ and $c$ aggregate message separately from **B** and **C**; 2. the central atom $a$ receives the aggregated message (which contains information of **B** and **C**) from its neighbors $b$ and $c$.

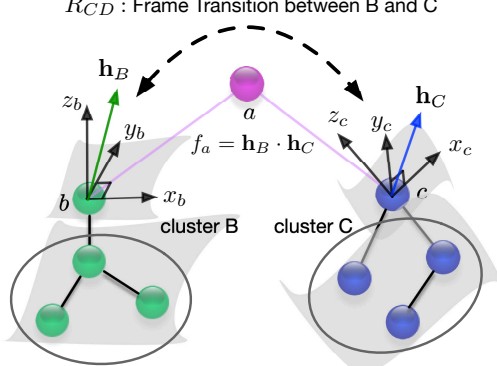

Figure 2: Illustrations of different local frames and their transition.

Let $S_{\mathbf{B}}$ ($S_{\mathbf{C}}$) denote the collection of all invariant scalars created by **B** (**C**) . For example, $S_{\mathbf{B}}$ contains all relative distances and angles within the 3D structure **B**. Then, the following theorem holds:

**Theorem 4.1.** *Not all types of invariant interaction $f_a(\boldsymbol{B}, \boldsymbol{C})$ can be expressed solely as functions of the union of two sets $S_{\boldsymbol{B}}$ and $S_{\boldsymbol{C}}$. In other words, there exists $E(3)$ invariant function $f_a(\boldsymbol{B}, \boldsymbol{C})$, such that it cannot be expressed as functions of $S_{\boldsymbol{B}}$ and $S_{\boldsymbol{C}}$: $f_a(\boldsymbol{B}, \boldsymbol{C}) \neq \rho(S_{\boldsymbol{B}}, S_{\boldsymbol{C}})$ for arbitrary invariant functions $\rho$.*

*Proof.* The detailed proof is provided in Appendix E. □

This theorem essentially demonstrates that simply aggregating "local" scalar information from different clusters is insufficient for approximating "global" interactions, even in the case of simple **invariant** potential learning tasks. In contrast to the previous section, where the local expressiveness was evaluated based on the ability to classify geometric shapes, in this theorem, we constructed counterexamples using continuous regression functions that depend strictly on information beyond the combination of local invariant scalars. Intuitively, the theorem is proved by introducing two local **equivariant frames** (three independent equivariant vectors, see Appendix B for the definition) determined by **B** (**C**) separately, through which all scalars in $S_{\mathbf{B}}$ ($S_{\mathbf{C}}$) can be expressed. However, the transition matrix between these frames is not encoded in the aggregation, resulting in **information loss** when aggregating geometric features from two sub-clusters. In other words, local frames provide local observations of geometric quantities, while the transition matrix reveals the global changes between local geometries. Importantly, the proof also highlights that the missing information that causes the expressiveness gap is solely the Frame Transition (FT) information, which we will define immediately.

**Frame Transition (FT).** Let $(e_1^i, e_2^i, e_3^i)$ and $(e_1^j, e_2^j, e_3^j)$ be two orthonormal frames in $\mathbf{R}^3$. These frames are connected by an orthogonal matrix $R_{ij} \in SO(3)$:

$$(\mathbf{e}_1^i, \mathbf{e}_2^i, \mathbf{e}_3^i) = R_{ij}(\mathbf{e}_1^j, \mathbf{e}_2^j, \mathbf{e}_3^j). \tag{5}$$

Furthermore, when $(e_1^i, e_2^i, e_3^i)$ and $(e_1^j, e_2^j, e_3^j)$ are equivariant frames, all elements of $R_{ij}$ are invariant scalars. For instance, in the case of a geometric graph where $i$ and $j$ represent indexes of two connected atoms, the fundamental torsion angle $\tau_{ij}$ in ComeNet [23] corresponds to one element of $R_{ij}$ (see Appendix E).

To address this expressiveness gap, one approach is to directly incorporate all invariant edge-wise frame transition matrices (**FT**) into the model. This can be achieved using a geometric formulation based on **neural sheaf diffusion**, as described in Appendix H. However, it is worth noting that this method becomes computationally expensive when dealing with a large number of local clusters, as it requires $O(k^2)$ pairs of **FT** for each node. Instead, we propose a more efficient approach by introducing equivariant tensor features for each node $i$, denoted as $\mathbf{m}_i$. These tensor features allow us to maintain equivariant frames directly within each $\mathbf{m}_i$. We prove in Appendix E that **FT** can be easily derived through equivariant message passing and updating when utilizing these equivariant tensor features.

**Equivariant Message Passing.** Similarly with the standard one-hop message passing scheme 1, the aggregated tensor message $\mathbf{m}_i$ from the $l-1$ layer to the $l$ layer can be written as: $\mathbf{m}_i^{l-1} = \sum_{j \in N(i)} \mathbf{m}_j^{l-1}$. Since summation does not break the symmetry rule, it is obvious that $\mathbf{m}_i^{l-1}$ are still equivariant tensors. However, the nontrivial part lies in the design of the **equivariant update** function $\phi$:

$$\mathbf{m}_i^l = \phi(\mathbf{m}_i^{l-1}). \tag{6}$$

To fully capture the information of **FT**, it is necessary for $\phi$ to possess sufficient expressive power while maintaining $SE(3)$ equivariance. Here, we propose a novel way of updating scalar and tensor messages by performing node-wise scalarization and tensorization blocks (the **FTE** module of Figure 3). From the perspective of Eq. 2, $\mathbf{m}(\mathbf{x}_u)$ is transformed equivariantly as:

$$\mathbf{m}(g\mathbf{x}_u) = \sum_{i=0}^{l} \mathcal{M}^i(g)\mathbf{m}_i(\mathbf{x}_u), \quad g \in SE(3). \tag{7}$$

Here, $\mathbf{m}(\mathbf{x}_u)$ is decomposed to $(\mathbf{m}_0(\mathbf{x}_u), \ldots, \mathbf{m}_l(\mathbf{x}_u))$ according to different tensor types, and $\{\mathcal{M}^i(g)\}_{i=0}^{l}$ is a collection of different $SE(3)$ **tensor representations** (see the precise definition in Appendix A).

To illustrate the benefit of aggregating equivariant messages from local patches, we study a simple case. Let $f_a(\mathbf{B}, \mathbf{C}) = \mathbf{h}_B \cdot \mathbf{h}_C$ be an invariant function of $\mathbf{B}$ and $\mathbf{C}$ (see Fig. 2), then $f_a$ can be calculated by a direction composition of scalar messages and equivariant vector messages: $f_a(\mathbf{B}, \mathbf{C}) = \frac{1}{2}[\|\mathbf{m}_a\|^2 - \|\mathbf{h}_B\|^2 - \|\mathbf{h}_C\|^2]$, where $\mathbf{m}_a = \mathbf{h}_B + \mathbf{h}_C$ is an equivariant vector. Note that $\mathbf{m}_a$ follows the local equivariant aggregation formula 6, and the other vectors' norm $\|\mathbf{h}_B\|$ and $\|\mathbf{h}_C\|$ are obtained through local scalarization on atoms $b$ and $c$. As a comparison, it's worth mentioning that $f_a(\mathbf{B}, \mathbf{C})$ can also be expressed by local scalarization with the additional transition matrix data $R_{BC}$ defined by Eq. 5. Let $\tilde{h}_B$ and $\tilde{h}_C$ be the scalarized coordinates with respect to two local equivariant frames $\mathcal{F}_B$ and $\mathcal{F}_C$. Then $f_a(\mathbf{B}, \mathbf{C}) = \frac{1}{2}\left[\left\|R_{BC}^{-1}\tilde{h}_B + \tilde{h}_C\right\|^2 - \left\|\tilde{h}_B\right\|^2 - \left\|\tilde{h}_C\right\|^2\right]$. However, it requires adding the transition matrix $R_{BC}$ for each $(\mathbf{B}, \mathbf{C})$ pair into the aggregation procedure, which is computationally expensive compared to directly implementing equivariant tensor updates.

## 5 Building an efficient and expressive equivariant 3D GNN

We propose to leverage the full power of **LSE** and **FTE** modules (along with a necessary **tensor update** module) to push the limit of efficient and expressive 3D equivariant GNNs design. Specifically, we improve a recently proposed method based on constructing local frames but without the implementation of **LSE** and **FTE** [30]. We also analyze related works following this framework detailed in Appendix G.

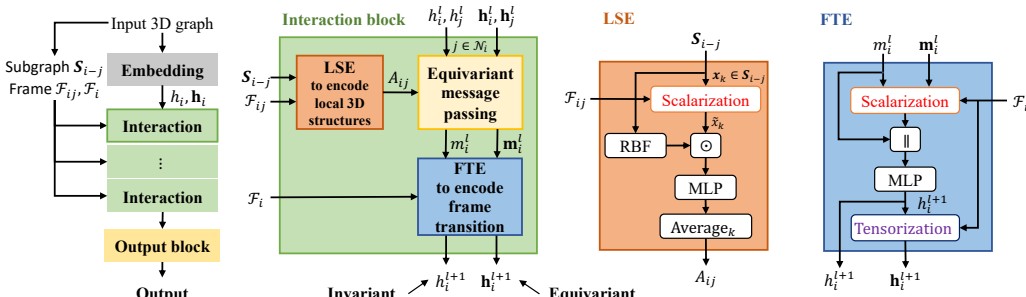

Figure 3: Illustrations of our modular framework for building equivariant GNNs and the realization of LEFTNet. Each interaction block contains **LSE** to encode local 3D structures, equivariant message passing to update both invariant (unbold letters, e.g. $h_i$) and equivariant (**bold** letter, e.g. $\mathbf{h}_i$) features, and **FTE** to encode frame transition. $\mathbf{S}_{i-j}$ is the local 3D structure of each edge $e_{ij}$. $\mathcal{F}_{ij}$ and $\mathcal{F}_i$ are the equivariant frames for each edge $e_{ij}$ and node $i$. $\odot$ indicates element-wise multiplication, and $\|$ indicates concatenation. Note that we do not include $\mathbf{e}_{ij}$ in the figure since, practically, they are generated based on $\mathbf{h}_i$ and $\mathbf{h}_j$.

**LSE Instantiation.** We propose to apply edge-wise equivariant frames (following [30]) to encode the local 3D structures $\mathbf{S}_{i-j}$. By definition, $\mathbf{S}_{i-j}$ contains edge $e_{ij}$, nodes $i$ and $j$, and their common neighbors. We use the equivariant frame $\mathcal{F}_{ij}$ built on $e_{ij}$ (see the precise formula in Appendix F) to scalarize $\mathbf{S}_{i-j}$. After scalarization (8), the equivariant coordinates of all nodes in $\mathbf{S}_{i-j}$ are transformed into invariant coordinates: $\{\mathbf{x}_k \to \tilde{x}_k \text{ for } \mathbf{x}_k \in \mathbf{S}_{i-j}\}$. To encode these scalars sufficiently, we first weight each $\tilde{x}_k$ by the **RBF** distance embedding: $\tilde{x}_k \to \mathbf{RBF}(\|\mathbf{x}_k\|) \odot \text{MLP}(\tilde{x}_k)$ for each $\mathbf{x}_k \in \mathbf{S}_{i-j}$. Note that to preserve the permutation symmetry, the MLP is shared among the nodes. Finally, the 3D structure weight $A_{ij}$ is obtained by the average pooling of all node features.

**FTE Instantiation.** We propose to introduce equivariant tensor message passing and update function for encoding local **FT** information. At initialization, let $\mathbf{NF}^l(\mathbf{x}_i, \mathbf{x}_j)$ denote the embedded tensor-valued edge feature between $i$ and $j$. We split it into two parts: 1. the scalar part $\text{SF}^l(\mathbf{x}_i, \mathbf{x}_j)$ for aggregating invariant messages; 2. the higher order tensor part $\mathbf{TF}^l(\mathbf{x}_i, \mathbf{x}_j)$ for aggregating tensor messages. To transform $\mathbf{TF}^l(\mathbf{x}_i, \mathbf{x}_j)$, we turn to the equivariant frame $\mathcal{F}_{ij}$ once again. After scalarization by $\mathcal{F}_{ij}$, $\mathbf{TF}^l(\mathbf{x}_i, \mathbf{x}_j)$ becomes a tuple of scalars $\tilde{\text{TF}}^l(\mathbf{x}_i, \mathbf{x}_j)$, which is then transformed by MLP. Finally, we output arbitrary tensor messages through equivariant tensorization 22:

$$\tilde{\text{TF}}^l(\mathbf{x}_i, \mathbf{x}_j) \xrightarrow[\mathcal{F}_{ij}]{\text{Tensorize}} \mathbf{NF}^{l+1}(\mathbf{x}_i, \mathbf{x}_j).$$

Further details are provided in Appendix F. As we have discussed earlier, the node-wise update function $\phi$ in Eq. 6 is also one of the guarantees for a powerful **FTE**. As a comparison, $\phi$ is usually a standard MLP for updating node features in 2D GNNs, which is a **universal approximator** of invariant functions. Previous works [14, 31] updated equivariant features by taking linear combinations and calculating the invariant norm of tensors, which may suffer from information loss. Then a natural question arises: *Can we design an equivariant universal approximator for tensor update?* We answer this question by introducing a novel node-wise frame. Consider node $i$ with its position $\mathbf{x}_i$, let $\bar{\mathbf{x}}_i := \frac{1}{N} \sum_{\mathbf{x}_j \in N(\mathbf{x}_i)} \mathbf{x}_j$ be the center of mass around $\mathbf{x}_i$'s neighborhood. Then the orthonormal equivariant frame $\mathcal{F}_i := (\mathbf{e}_1^i, \mathbf{e}_2^i, \mathbf{e}_3^i)$ with respect to $\mathbf{x}_i$ is defined similar to the edge-wise frame between $\mathbf{x}_i$ and $\bar{\mathbf{x}}_i$ in [30], detailed in appendix B. Finally, we realize a powerful $\phi$ by the following theorem:

**Theorem 5.1.** *Equipped with an equivariant frame $\mathcal{F}_i$ for each node $i$, the equivariant function $\phi$ defined by the following composition is a universal approximator of tensor transformations:*
$\phi$ : ***Scalarization*** $\to$ ***MLP*** $\to$ ***Tensorization***.

*Proof.* The detailed proof is provided in Appendix F. □

**LEFTNet.** An overview of our $\{\mathbf{LSE}, \mathbf{FTE}\}$ enhanced efficient graph neural network (LEFTNet) is depicted in Fig. 3. The detailed algorithm for LEFTNet is shown in Algorithm 1 of Appendix C. LEFTNet receives as input a collection of node embeddings $\{v_1^0, \ldots, v_N^0\}$, which contain the atom types and 3D positions for each node: $v_i^0 = (z_i, \mathbf{x}_i)$, where $i \in \{1, \ldots, N\}$. For each edge $e_{ij} \in E$,

we denote the associated equivariant features consisting of tensors by $\mathbf{e}_{ij}$. During each messaging passing layer, the **LSE** module outputs the scalar weight coefficients $A_{ij}$ as enhanced invariant edge feature and feed into the interaction module. Moreover, scalarization and tensorization as two essential blocks are used in the equivariant update module that fulfills the function of **FTE**. The permutation equivariance of a geometric graph is automatically guaranteed for any message passing architecture, we provide a complete proof of SE(3)-equivariance for LEFTNet in Appendix F.

**SE(3) vs E(3) Equivariance.** Besides explicitly fitting the $SE(3)$ invariant molecular geometry probability distribution, modeling the energy surface of a molecule system is also a crucial task for molecule property prediction. However, the Hamiltonian energy function $E$ of a molecule is invariant under refection transformation: **Energy**$(\mathbf{X})$ = **Energy**$(R\mathbf{X})$, for arbitrary reflection transformation $R \in E(3)$. In summary, there exist two different inductive biases for modeling 3D data: **(1)** SE(3) equivariance, e.g. chirality could turn a therapeutic drug to a killing toxin; **(2)** E(3) equivariance, e.g. energy remains the same under reflections.

Since we implement $SE(3)$ equivariant frames in LEFTNet, our algorithm is naturally $SE(3)$ equivariant. However, our method is **flexible** to implement $E(3)$ equivariant tasks as well. For $E(3)$ equivariance, we can either replace our frames to $E(3)$ equivariant frames, or modify the scalarization block by taking the absolute value: $\mathbf{x} \rightarrow \tilde{x} := \underbrace{(\mathbf{x} \cdot e_1, \mathbf{x} \cdot e_2, \mathbf{x} \cdot e_3)}_{SE(3)} \rightarrow \underbrace{(\mathbf{x} \cdot e_1, |\mathbf{x} \cdot e_2|, \mathbf{x} \cdot e_3)}_{E(3)}$.

Intuitively, since the second vector $e_2$ is a pseudo-vector, projections of any equivariant vectors along the $e_2$ direction are not $E(3)$ invariant until taking the absolute value.

**Efficiency.** To analyze the efficiency of LEFTNet, suppose 3D graph $G$ has $n$ vertices, and its average node degree is $k$. Our algorithm consists of three phases: 1. Building equivariant frames and performing local scalarization; 2. Equivariant message passing; 3. Updating node-wise tensor features through scalarization and tensorization. Let $l$ be the number of layers, then the computational complexity for each of our three phases are: 1. $O(nk)$ for computing the frame and local (1-hop) 3D features; 2. $O(nkl)$ for 1-hop neighborhood message aggregation; 3. $O(nl)$ for node-wise tensorization and feature update.

## 6 Experiments

We evaluate our LEFTNet on both scalar value (e.g. energy) and vector value (e.g. forces) prediction tasks. The scalar value prediction experiment is conducted on the QM9 dataset [32] which includes $134k$ small molecules with quantum property annotations; the vector value prediction experiment is conducted on the MD17 dataset [33] and the Revised MD17(rMD17) dataset [34] which includes the energies and forces of molecules. We compare LEFTNet with a list of state-of-the-art equivariant (invariant) GNNs including SphereNet [15], PaiNN [35], Equiformer [36], GemNet [37], etc [28, 38, 13, 39, 40, 14, 16, 41–48].The training details, results on rMD17, etc. are listed in Appendix I.

Table 1: Mean Absolute Error for the molecular property prediction benchmark on QM9 dataset. (The best results are **bolded** and the second best are underlined.)

| Task | $\alpha$ | $\Delta\varepsilon$ | $\varepsilon_{\mathrm{HOMO}}$ | $\varepsilon_{\mathrm{LUMO}}$ | $\mu$ | $C_\nu$ | $G$ | $H$ | $R^2$ | $U$ | $U_0$ | ZPVE |
| Units | bohr$^3$ | meV | meV | meV | D | cal/mol K | meV | meV | bohr$^3$ | meV | meV | meV |
|---|---|---|---|---|---|---|---|---|---|---|---|---|
| NMP | .092 | 69 | 43 | 38 | .030 | .040 | 19 | 17 | .180 | 20 | 20 | 1.50 |
| Cormorant | .085 | 61 | 34 | 38 | .038 | .026 | 20 | 21 | .961 | 21 | 22 | 2.03 |
| LieConv | .084 | 49 | 30 | 25 | .032 | .038 | 22 | 24 | .800 | 19 | 19 | 2.28 |
| TFN | .223 | 58 | 40 | 38 | .064 | .101 | - | - | - | - | - | - |
| SE(3)-Tr. | .142 | 53 | 35 | 33 | .051 | .054 | - | - | - | - | - | - |
| EGNN | .071 | 48 | 29 | 25 | .029 | .031 | 12 | 12 | **.106** | 12 | 11 | 1.55 |
| SEGNN | .060 | 42 | 24 | 21 | .023 | .031 | 15 | 16 | .660 | 13 | 15 | 1.62 |
| ClofNet | .063 | 53 | 33 | 25 | .040 | .027 | 9 | 9 | .610 | 9 | 8 | **1.23** |
| EQGAT | .063 | 44 | 26 | 22 | .014 | .027 | 12 | 13 | .257 | 13 | 13 | 1.50 |
| Equiformer | .056 | **33** | **17** | **16** | .014 | .025 | 10 | 10 | .227 | 11 | 10 | 1.32 |
| LEFTNet (ours) | **.048** | 40 | 24 | 18 | **.012** | **.023** | 7 | 6 | .109 | 7 | 6 | 1.33 |
| Schnet | .235 | 63 | 41 | 34 | .033 | .033 | 14 | 14 | .073 | 19 | 14 | 1.70 |
| DimeNet++ | .044 | 33 | 25 | 20 | .030 | .023 | 8 | 7 | .331 | 6 | 6 | 1.21 |
| SphereNet | .046 | **32** | **23** | **18** | .026 | **.021** | 8 | 6 | .292 | 7 | 6 | **1.12** |
| ClofNet | .053 | 49 | 33 | 25 | .038 | .026 | 9 | 8 | .425 | 8 | 8 | 1.59 |
| PaiNN | .045 | 46 | 28 | 20 | .012 | .024 | 7 | 6 | **.066** | 6 | 6 | 1.28 |
| LEFTNet (ours) | .039 | 39 | **23** | **18** | **.011** | .022 | **6** | **5** | .094 | 5 | 5 | 1.19 |

Table 2: Mean Absolute Error for per-atom forces prediction (kcal/mol Å) on MD17 dataset. The best results are **bolded**.

| Molecule | WoFE=100 | | | | | | WoFE=1000 | | | Others | |
| --- | --- | --- | --- | --- | --- | --- | --- | --- | --- | --- | --- |
| | sGDML | SchNet | DimeNet | SphereNet | SpookyNet | LEFTNet | SphereNet | GemNet | LEFTNet | PaiNN | NewtonNet |
| Aspirin | 0.68 | 1.35 | 0.499 | 0.430 | **0.258** | 0.300 | 0.209 | 0.217 | **0.196** | 0.371 | 0.348 |
| Benzene | 0.20 | 0.31 | 0.187 | 0.178 | – | **0.145** | 0.147 | 0.145 | **0.142** | – | – |
| Ethanol | 0.33 | 0.39 | 0.230 | 0.208 | **0.094** | 0.138 | 0.091 | **0.086** | 0.099 | 0.230 | 0.264 |
| Malonaldehyde | 0.41 | 0.66 | 0.383 | 0.340 | **0.167** | 0.209 | 0.172 | 0.155 | **0.142** | 0.319 | 0.323 |
| Naphthalene | 0.11 | 0.58 | 0.215 | 0.178 | 0.089 | **0.073** | 0.048 | 0.051 | **0.044** | 0.083 | 0.084 |
| Salicylic acid | 0.28 | 0.85 | 0.374 | 0.360 | 0.180 | **0.167** | **0.113** | 0.125 | 0.117 | 0.209 | 0.197 |
| Toluene | 0.14 | 0.57 | 0.210 | 0.155 | 0.087 | **0.084** | 0.054 | 0.060 | **0.049** | 0.102 | 0.088 |
| Uracil | 0.24 | 0.56 | 0.301 | 0.267 | 0.119 | **0.116** | 0.106 | 0.097 | **0.085** | 0.140 | 0.149 |

## 6.1 QM9 - scalar-valued property prediction

The QM9 dataset is a widely used dataset for predicting molecular properties. However, existing models are trained on different data splits. Specifically, Cormorant [40], EGNN [14], etc., use 100k, 18k, and 13k molecules for training, validation, and testing, while DimeNet [38], SphereNet [15], etc., split the data into 110k, 10k, and 11k. For a fair comparison with all methods, we conduct experiments using both data splits. Results are listed in Table 1. For the first data split, LEFTNet is the best on 7 out of the 12 properties and improves previous SOTA results by 20% on average. Consistently, LEFTNet is the best or second best on 10 out of the 12 properties for the second split. These results validate the effectiveness of LEFTNet. Detailed ablation in Appendix I shows that both **LSE** and **FTE** contribute to the final performance. In addition, we compare the forward time of existing methods to show the efficiency of LEFTNet. Specifically, we use the same batch size for all methods and report the one epoch training time. Fig. 4 shows that LEFTNet can achieve the best performance using similar forward time as SchNet, ClofNet, and ComENet and is much faster than DimeNet and SphereNet. This is consistent with our efficiency analysis in Sec. 5 and Table 4.

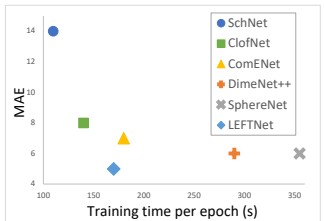

Figure 4: Comparisons of existing methods in terms of the training time and the MAE of the property $U_0$ of QM9.

## 6.2 MD17 - vector-valued property prediction

We evaluate LEFTNet to predict forces on the MD17 dataset. Following existing studies [28, 38, 15], we train a separate model for each of the 8 molecules. Both training and validation sets contain 1000 samples, and the rest are used for testing. Note that all baseline methods are trained on a joint loss of energies and forces, but different methods use different weights of force over energy (WoFE). For example, SchNet [28] sets WoEF as 100, while GemNet [37] uses a weight of 1000. For a fair comparison with existing studies, we conduct experiments on two widely used weights of 100 and 1000 following Liu et al. [15]. The results are summarized in Table 2. Results show that when WoFE is 100, LEFTNet outperforms all baseline methods on 5 of the 8 molecules. In addition, LEFTNet can outperform all baseline methods on 6 of the 8 molecules when WoFE is 1000. These experimental results on MD17 demonstrate the performance of LEFTNet on vector-valued property prediction tasks. The ablation results in Table 3 demonstrate that both **LSE** and **FTE** are important to the final results. Specifically, the algorithm for **LSE** only is in Algorithm 1 of Appendix C. Note that the original MD17 dataset we used has numerical noise [34], and a recomputed version of MD17, called Revised MD17 (rMD17) [34] is proposed to reduce the numerical noise. In addition to the original MD17, we also conduct experiments on the rMD17, and the results are listed in Appendix I.

Table 3: Ablation study on Aspirin of the MD17 dataset. Detailed ablations on other molecules are listed in Appendix I

| Method | MAE |
| --- | --- |
| w/o **LSE** and **FTE** | 1.083 |
| **LSE** only | 0.451 |
| **LSE** + **FTE** | **0.300** |

## 7 Limitation and future work

In this paper, we seek a general recipe for building 3D geometric graph deep learning algorithms. Considering common prior of 2D graphs, such as permutation symmetry, has been incorporated in off-the-shelf graph neural networks, we mainly focus on the $E(3)$ and $SE(3)$ symmetry specific to 3D geometric graphs. Despite our framework being general for modeling geometric objects, we only conducted experiments on commonly used molecular datasets. It's worth exploring datasets in other domains in the future.

To elucidate the future design space of equivariant GNNs, we propose two directions that are worth exploring. Firstly, our current algorithms consider fixed equivariant frames for performing aggregation and node updates. Inspired by the high body-order ACE approach [49] (for modeling atom-centered potentials), it is worth investigating in the future if equivariant frames that relate to many body (e.g., the PCA frame in [50, 51]) can boost the performance of our algorithm. For example, to build the A-basis proposed in [49], we can replace our message aggregation Eq. 6 from summation to tensor product, which is also a valid pooling operation. Another direction is to explore geometric mesh graphs on manifolds $M$, where the local frame is defined on the tangent space of each point: $\mathcal{F}(x) \in T_x M$. Since our scalarization technique (crucial for realizing **LSE** in LEFTNet) originates from differential geometry on frame bundles [52], it is reasonable to expect that our framework also works for manifold data [53, 54].

## Acknowledgments and Disclosure of Funding

Y.D., D.F. and C.P.G. are supported by the Eric and Wendy Schmidt AI in Science Postdoctoral Fellowship, a Schmidt Futures program; the National Science Foundation (NSF), the Air Force Office of Scientific Research (AFOSR). L.W. and S.J. are supported by NSF IIS-2243850.

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

# Appendix for "A new perspective on building efficient and expressive 3D equivariant graph neural networks"

## Contents

## A  Supplementary background

In this section, we first detail the concept of scalarization and tensorization and then review the concept of (contravariant) tensor fields and their associated equivariant group representations.

**Scalarization.** Scalarization is a general technique that originated from differential geometry for realizing covariant operations on tensors [52]. Our method will apply a simple version of scalarization in $\mathbf{R}^3$ to transform equivariant quantities. At the heart of its realization is the notion of equivariant orthonormal frames, which consist of three orthonormal equivariant vectors:

$$\mathcal{F} := (\mathbf{e}_1, \mathbf{e}_2, \mathbf{e}_3).$$

Based on $\mathcal{F}$, we can build orthonormal equivariant frames for higher order tensors by taking tensor products $\otimes$, see Eq. 21 in Appendix. By taking the inner product between $\mathcal{F}$ and a given equivariant vector (tensor) $\mathbf{x}$, we get a tuple of invariant scalars (see [30] for a proof):

$$\mathbf{x} \to \tilde{x} := (\mathbf{x} \cdot \mathbf{e}_1, \mathbf{x} \cdot \mathbf{e}_2, \mathbf{x} \cdot \mathbf{e}_3), \tag{8}$$

and $\tilde{x}$ can be seen as the 'scalarized' coordinates of $\mathbf{x}$.

**Tensorization.** Tensorization, on the other hand, is the 'reverse' process of scalarization. Given a tuple of scalars: $(x_1, x_2, x_3)$, tensorization creates an equivariant vector (tensor) out of $\mathcal{F}$:

$$(x_1, x_2, x_3) \xrightarrow{\text{Pairing}} \mathbf{x} := x_1 \mathbf{e}_1 + x_2 \mathbf{e}_2 + x_3 \mathbf{e}_3. \tag{9}$$

The same procedure is extended to higher order cases, see Eq. 22 in Appendix.

A $s$ order (contravariant) tensor $\mathbf{T}$ on a vector space $\mathbf{V}$ is a multilinear map:

$$T : \underbrace{\mathbf{V}^* \times \cdots \times \mathbf{V}^*}_{s} \to \mathbf{R}^1,$$

where $\mathbf{V}^*$ denotes the dual space of $\mathbf{V}$. In fact, there is a canonical 'multiplication' operation between two tensors. Define the **tensor product** $\mathbf{S} \otimes \mathbf{T}$ of two tensors $\mathbf{S}$ and $\mathbf{T}$ to be a tensor of order $r + s$ :

$$\mathbf{S} \otimes \mathbf{T}(v^1, \ldots, v^{r+s}) = \mathbf{S}(v^1, \ldots, v^r)\mathbf{T}(v^{r+1}, \ldots, v^{r+s}). \tag{10}$$

where $v^i \in \mathbf{V}^*$.

From now on, we assume $V = V^* = \mathbf{R}^3$. Note that when $s = 1$, $\mathbf{T}$ is exactly an equivariant vector. In practice, the tensor data in $\mathbf{R}^3$ is usually given by its coefficients under a Cartesian coordinate system. Take a second-order tensor as an example, assume we are given an orthonormal frame (basis) $(\mathbf{e}_1, \mathbf{e}_2, \mathbf{e}_3)$ and its dual frame $(\mathbf{e}^1, \mathbf{e}^2, \mathbf{e}^3)$, then the nine coefficients of $T$ are given by

$$T_{ij} = \mathbf{T}(\mathbf{e}^i, \mathbf{e}^j), \ \ 1 \le i, j \le 3.$$

In other words, we say the collection $\{T_{ij}\}_{1 \le i,j \le 3}$ is a faithful representation of $\mathbf{T}$ in a fixed coordinate system:

$$\mathbf{T} = \sum_{i,j} T_{ij} \mathbf{e}_i \otimes \mathbf{e}_j. \tag{11}$$

Once defined a tensor on $\mathbf{R}^3$, it's easy to extend it to a continuous manifold or a discrete graph. A **tensor field** of order $s$ on a 3D graph $G = (V, E)$ is a tensor-valued function $f$ which assigns to each 3D node $\mathbf{x}_i$ an order $s$ tensor, denoted by $f(\mathbf{x}_i)$.

**SE(3) Tensor Representations.** Let $V$ be a vector space, then the group SE(3) is said to act on $V$ if there is a mapping $\phi : SE(3) \times V \to V$ satisfying the following two conditions:

1. if $e \in SE(3)$ is the identity element, then

$$\phi(e, x) = x \quad \text{for} \ \forall x \in V.$$

2. if $g_1, g_2 \in SE(3)$, then

$$\phi(g_1, \phi(g_2, x)) = \phi(g_1 g_2, x) \quad \text{for} \ \forall x \in V.$$

If we further require $\phi(g, \cdot)$ is a linear map for all $g \in SE(3)$, then $\phi$ becomes a group representation of $SE(3)$. From now on, we only consider the rotation subgroup SO(3) and its group representations. When $V = \mathbf{R}^3$, there is a natural representation of $SO(3)$ by rotating vectors in $\mathbf{R}^3$. In this way, an element $g \in SO(3)$ is identified with a Rotation matrix, denoted by $\{g_i^j\}_{1 \le i,j \le 3}$.

From the tensor definition (10), this natural representation on $\mathbf{R}^3$ induces a tensor representation on $T$. Still take $\mathbf{T} = \{T_{ij}\}_{1 \le i,j \le 3}$ as an example, we have

$$T_{kl} = \sum_i \sum_j g_k^i g_l^j T_{ij}, \quad 1 \le k, l \le 3, \tag{12}$$

for $\forall g \in SO(3)$. It's easy to check that (12) is indeed a SO(3) representation on the vector space spanned by second-order tensors.

**Relation with Spherical Harmonics.** For the $SO(3)$ group, all representations (including the tensor representations) can be decomposed as a direct sum of irreducible representations. For each type of irreducible representations, there is a subset of spherical harmonics formulating a basis for this specific representation. However, in terms of representing equivariant geometric quantities, the theorem in [55] claims that tensor representations and irreducible representations are equally powerful: They all form a complete basis in the space of continuous $E(3)$ equivariant functions.

## B  Equivariant frame construction

**Equivariant Frame Definition.** An equivariant frame on a 3D graph $G$ consists of three orthonormal vectors that transform equivariantly under $SE(3)$. In the following, we present specific constructions of equivariant frames. However, it is important to note that the proof of Theorem 4.1 does not depend on our particular choice of local equivariant frames.

It is worth noting that the number of vectors required in the equivariant frame is equal to the dimension of the space. For example, in the case of the four-dimensional Lorentz space, where the symmetry group is replaced by the Lorentz group, the equivariant frame consists of four independent vectors.

**Edge-wise frame.** From this point onward, we assume that the 3D graph's mass has been translated to zero. This translation ensures that the center of mass of the graph is located at the origin, ensuring the translation invariance of the system. Consider node $i$ and one of its neighbors $j$ with positions $\mathbf{x}_i$ and $\mathbf{x}_j$, respectively. The orthonormal equivariant frame $\mathcal{F}_{ij} := (\mathbf{e}_1^{ij}, \mathbf{e}_2^{ij}, \mathbf{e}_3^{ij})$ is defined with respect to $\mathbf{x}_i$ and $\mathbf{x}_j$ as follows:

$$\left( \frac{\mathbf{x}_i - \mathbf{x}_j}{\|\mathbf{x}_i - \mathbf{x}_j\|}, \frac{\mathbf{x}_i \times \mathbf{x}_j}{\|\mathbf{x}_i \times \mathbf{x}_j\|}, \frac{\mathbf{x}_i - \bar{\mathbf{x}}_j}{\|\mathbf{x}_i - \bar{\mathbf{x}}_j\|} \times \frac{\mathbf{x}_i \times \mathbf{x}_j}{\|\mathbf{x}_i \times \mathbf{x}_j\|} \right). \tag{13}$$

**Node-wise frame.** Consider node $i$ with 3D position $\mathbf{x}_i$, and let $\bar{\mathbf{x}}_i := \frac{1}{N} \sum_{\mathbf{x}_j \in \mathcal{N}(\mathbf{x}_i)} \mathbf{x}_j$ be the center of mass around the 1-hop neighborhood of $\mathbf{x}_i$. The orthonormal equivariant frame $\mathcal{F}_i := (\mathbf{e}_1^i, \mathbf{e}_2^i, \mathbf{e}_3^i)$ is defined with respect to $\mathbf{x}_i$ as follows:

$$\left( \frac{\mathbf{x}_i - \bar{\mathbf{x}}_i}{\|\mathbf{x}_i - \bar{\mathbf{x}}_i\|}, \frac{\bar{\mathbf{x}}_i \times \mathbf{x}_i}{\|\bar{\mathbf{x}}_i \times \mathbf{x}_i\|}, \frac{\mathbf{x}_i - \bar{\mathbf{x}}_i}{\|\mathbf{x}_i - \bar{\mathbf{x}}_i\|} \times \frac{\bar{\mathbf{x}}_i \times \mathbf{x}_i}{\|\bar{\mathbf{x}}_i \times \mathbf{x}_i\|} \right). \tag{14}$$

Note that node frames can also be obtained by averaging the edge-wise frames and applying the Gram-Schmidt orthogonalization process. This approach provides an alternative method for constructing equivariant frames at the node level.

## C  Algorithm for the proposed LEFTNet

The detailed LEFTNet is shown in Algorithm 1. Note that for simplicity, the message $\mathbf{m}$ in this algorithm could include both invariant and equivariant terms in Fig. 3.

The design of LEFTNet (**LSE** only) used in the ablation study is shown in Algorithm 2.

---

**Algorithm 1** LEFTNet

---

1: **Input:** 3D graph with equivariant positions $\mathbf{X} = (\mathbf{x}_1, \ldots, \mathbf{x}_n) \in \mathbb{R}^{n \times 3}$, invariant node features $h_i \in \mathbb{R}^d$, invariant relative distances $d_{ij} \in \mathbb{R}^1$, **equivariant** edge features $\mathbf{e}_{ij} \in \mathbb{R}^c$.
2: Centralize the positions: $\mathbf{X} \leftarrow \mathbf{X} - \text{CoM}(\mathbf{X})$.
3: **for** $(i = 1; i <= n; i + +)$ **do**
4:     Compute node-wise equivariant frames $\mathcal{F}_i$ via Eq. 14 .
5:     **for** $j \in \mathcal{N}(i)$ **do**
6:         Compute edge-wise equivariant frames $\mathcal{F}_{ij}$ via Eq. 13
7:         Get the mutual 3D structure $\mathbf{S}_{i-j}$, perform local scalarization through $\mathcal{F}_{ij}$:

$$t_{ij} = \{\mathbf{Scalarize}(\mathbf{S}_{i-j}, \mathcal{F}_{ij})\}$$

8:         Calculate the SE(3)-invariant structural coefficients: $A_{ij} = g(t_{ij}, d_{ij})$
9:         Perform equivariant message passing as in Eq. 4:

$$\mathbf{m}_{ij} = \phi_m^1(h_i, A_{ij} \odot h_j, d_{ij}) + \phi_m^2(h_i, A_{ij} \odot h_j, d_{ij}) \cdot \mathbf{e}_{ij} + \phi_m^3(h_i, A_{ij} \odot h_j, d_{ij}) \cdot \mathcal{F}_{ij}$$

10:     **end for**
11:     Equivariant message aggregation: $\mathbf{m}_i = \sum_{j \in \mathcal{N}(i)} \mathbf{m}_{ij}$;
12:     Transform equivariant node features through $\mathcal{F}_i$:

$$t_i = \mathbf{Scalarize}(\mathbf{m}_i, \mathcal{F}_i)$$

13:     Update invariant node features:

$$h_i = \phi_h(h_i, t_i)$$

14:     **Equivariant Output:** Perform tensorization through $\mathcal{F}_i$:

$$\mathbf{h}_i = \mathbf{Tensorize}(h_i, \mathcal{F}_i).$$

15: **end for**
16: **Output:** for invariant properties (e.g. energy): AvgPooling($h_1, \ldots, h_n$),
    for equivariant properties (e.g. force): $(\mathbf{h}_1, \ldots, \mathbf{h}_n)$.

---

**Algorithm 2** LEFTNet (**LSE** only)

---

1: **Input:** 3D graph with equivariant positions $\mathbf{X} = (\mathbf{x}_1, \ldots, \mathbf{x}_n) \in \mathbb{R}^{n \times 3}$, invariant node features $h_i \in \mathbb{R}^d$, invariant relative distances $d_{ij} \in \mathbb{R}^1$.
2: Centralize the positions: $\mathbf{X} \leftarrow \mathbf{X} - \text{CoM}(\mathbf{X})$.
3: **for** $(i = 1; i <= n; i++)$ **do**
4:    **for** $j \in \mathcal{N}(i)$ **do**
5:       Compute edge-wise equivariant frames $\mathcal{F}_{ij}$ via Eq. 13:

$$\mathcal{F}_{ij} = \textbf{EquiFrame}(\mathbf{x}_i, \mathbf{x}_j).$$

6:       Get the mutual 3D structure $\mathbf{S}_{i-j}$, perform local scalarization:

$$t_{ij} = \{\textbf{Scalarize}(\mathbf{S}_{i-j}, \mathcal{F}_{ij})\}.$$

7:       Calculate the SE(3)-invariant structural coefficients: $A_{ij} = g(t_{ij}, d_{ij})$
8:       Perform invariant message passing:

$$m_{ij} = \phi_m(h_i, A_{ij} \odot h_j, d_{ij})$$

9:    **end for**
10:   Update invariant node features:

$$h_i = \phi_h(h_i, \sum_{j \in \mathcal{N}(i)} m_{ij})$$

11: **end for**
12: **Output: AvgPooling** $(h_1, \ldots, h_n)$.

---

# D   Related proofs and discussions of Section 3

In section 3, we proposed a novel hierarchy of local geometric isomorphisms that further motivates the design of incorporating the mutual 3D substructure's information into equivariant GNNs. Different from our fine-grained local characterization, a cocurrent work GWL [22]) proposes to measure geometric isomorphism from local to global by the k-hop partition.

From another point of view, we essentially demonstrated that encoding mutual 3D substructures expands the capacity of the transformation function class with respect to an equivariant GNN. [23] put forward the **Completeness** concept for characterizing these transformation functions. However, it mainly concentrates on testing whether a function can discriminate global geometric isomorphism (in the sense of Eq. 3).

**Discussion on the Completeness Concept.** Following our terminology in the preliminary section, **completeness** of a transformation $f$ can be translated into claiming that $f$ is invariant among 3D graphs if and only if they are **globally** isomorphic (see definition (3)). Therefore, it's easy to refine the notion of completeness that adapts to our local version by replacing the global isomorphism to local isomorphism:

$$f(\mathbf{X}) = f(\mathbf{Y}),$$

if and only if $\mathbf{X}$ and $\mathbf{Y}$ are local {tree-, triangular-, subgraph-} isomorphic. Then, in terms of function class capacities, the following relation holds:

**Global complete** $\subset$ **Subgraph complete** $\subset$ **Triangular complete** $\subset$ **Tree complete**.

Note that our equivalent description of complete transformation reveals the fact that the completeness concept in [23] is defined from the global 3D isomorphism point of view. Therefore, we shall claim that the above series of completeness notions belong to the **structure** completeness. Indeed, the theory developed in section 3 indicates that a GNN which can express **structure** complete functions may not be sufficient in expressing general tensor potential functions on a 3D graph.

On the other hand, a non-negligible proportion of 3D graph tasks may not be sensitive to the global 3D non-isomorphism. For example, some chemical properties (formulated as a function defined on molecular graphs) are characterized by local substructures [56]. In these scenarios, we are looking for a geometric transformation $f$ that is global non-complete, but (tree-, triangular-, subgraph-) local complete.

**Proof of Theorem 3.1.**

**Remark.**

1. When compared to the construction of topological structural coefficients that take the discrete adjacency matrix of the graph as input, as proposed in [27], the input 3D structural coefficients we seek are of a continuous nature. For instance, as illustrated in Fig. 1, the key features that differentiate these 3D local shapes are the dihedral angle and relative distance, both of which are continuous functions of the 3D positions. Therefore, we require the existence of universal approximators for continuous functions to ensure that the parametrization of $A_{ij}$ has sufficient expressiveness.

2. The reason we desire the **LSE** module to be invariant is because it ensures the local injectivity of message passing, following the standard argument proposed in Xu et al. [25]. This local injectivity is crucial for proving the second part of Theorem 3.1.

3. From a mathematical perspective, the common approach to proving the universal approximator property is by demonstrating that a neural network can express a complete basis in the function space, typically leveraging the Stone-Weierstrass theorem. Two successful examples are: a. Piecewise linear function bases achieved by wide Multi-layer perceptrons (**MLP**); b. Polynomial bases. In the 3D GNN setting, equivariant polynomials that are also permutation-invariant can be rigorously constructed using ACE [29]. Additionally, through local scalarization and the Kolmogorov representation theorem [57], the universal approximator property of 3D GNNs reduces to the universal approximator property of **MLP**, which has been proven for a broad range of non-linear activation functions.

*Proof.* The first part of the theorem is proved by providing an explicit example. Consider the first pair of 3D shapes in Figure 1, where the mutual 3D substructures $\mathbf{S}_{i-q}$ and $\mathbf{S}_{j-q'}$ both contain two triangles: $(x_i, x_p, x_q)$, $(x_i, x_m, x_q)$ and $(x_j, x_{p'}, x_{q'})$, $(x_j, x_{m'}, x_{q'})$ respectively.

The difference between these two triangular non-isomorphic shapes can be captured by the distance function:

$$d(\mathbf{x}_p, \mathbf{x}_m) = \|\mathbf{x}_p - \mathbf{x}_m\|^2 \,,$$

and similarly, $d(\mathbf{x}p', \mathbf{x}m')$ for the second shape. Note that this function cannot be expressed solely by tree-level features since there is no edge connecting $\mathbf{x}_p$ ($\mathbf{x}_{p'}$) and $\mathbf{x}_m$ ($\mathbf{x}_{m'}$). The fact that this function produces different output values for the two tree isometric but triangular non-isometric 3D shapes implies that $\phi$ is capable of distinguishing 3D shapes beyond tree isomorphism.

Therefore, if the structural coefficients $A_{ip}$ ($A_{jp'}$) take the value of $d(\mathbf{x}_p, \mathbf{x}_m)$ ($d(\mathbf{x}_{p'}, \mathbf{x}_{m'})$), then we know that it can differentiate these two triangular non-isomorphic shapes. Thus, it suffices to prove that the encoder $\phi$ is a universal approximator of continuous functions (which obviously contain the distance functions) that takes the mutual 3D substructures $\mathbf{S}_{i-q}$ ($\mathbf{S}_{j-q'}$) as input.

According to the Stone-Weierstrass theorem, it is sufficient to demonstrate that $\phi$ can express a complete basis within the continuous function space. In the 3D GNN setting, this basis should satisfy two additional requirements: 1) permutation equivariance, and 2) $SE(3)$ equivariance. An example of such a basis is the set of $E(3)$ equivariant polynomials that are also permutation equivariant, as demonstrated in Drautz [29]. To obtain invariant coefficients from the combination of equivariant basis functions, we can employ a projection layer, similar to the one used in Dym and Maron [55].

Alternatively, a simpler approach to achieve the universal approximator property is by leveraging local scalarization. Specifically, we use an equivariant frame on the edge $e_{iq}$ (as described in the previous section) to scalarize the mutual 3D structure $\mathbf{S}_{i-q}$ ($\mathbf{S}_{j-q'}$), which includes $\mathbf{x}_p$ ($\mathbf{x}_{p'}$) and $\mathbf{x}_m$ ($\mathbf{x}_{m'}$). Let $\tilde{S}_{i-q}$ denote the scalarized 3D positions of $\mathbf{S}_{i-q}$. By quoting the information lossless theorem and the Kolmogorov representation theorem for permutation-invariant functions from Zaheer et al. [57], Du et al. [30], we know that any $\varphi$ that is a $SE(3)$ and permutation invariant function of $\mathbf{S}_{i-q}$ can be expressed as:

$$\varphi(\mathbf{S}_{i-q}) = f\left( \sum_{x \in \tilde{S}_{i-q}} g(x) \right). \tag{15}$$

where $f$ and $g$ are certain continuous functions that can be approximated by wide **MLP**, following the universal approximator property of **MLP** [25].

The second part of the proof demonstrates that there exists an enhanced message passing framework (under injectivity assumptions, which can be realized similarly to [25, 27]) that incorporates the structural coefficients $\{\{A_{ij} := \phi(\mathbf{S}_{i-j})\}_{e_{ij} \in E}\}$ to map two distinct local 3D subgraphs with isometric local tree structures in Figure 1 to different embeddings. Since we specifically construct $\phi$ as an invariant encoder, the collection $\{\{A_{ij} := \phi(\mathbf{S}_{i-j})\}_{e_{ij} \in E}\}$ consists of invariant numbers, and the construction of the message passing framework follows a similar approach to [27]. For completeness, we provide a brief review of the injectivity condition and the message passing construction.

To establish the injectivity condition and prove the second part, we introduce the multi-set notation $\{\{\cdot\}\}$, following Xu et al. [25]. A basic equivariant GNN within our enhanced framework consists of at least two steps: 1. Message passing, defined by (4); 2. Node-wise update:

$$h_i^{t+1} = \mathbf{MLP}(m_i^t, h_i^t).$$

For simplicity, we denote the composition of these two steps as $\Psi$. Then, the additional injectivity condition requires that $\Psi$ satisfies the following:

$$\Psi(\{\{h_i^t, A_{ji}h_i^t, , h_j^t | j \in \mathcal{N}_i\}\}, \{\{A_{ij}h_j^t | j \in \mathcal{N}_i\}\}) \tag{16}$$

is injective for each layer $t$ and each node $i$. It is worth noting that this condition can be trivially achieved by incorporating weighted residue terms, similar to formula (5) in Wijesinghe and Wang [27]. Consequently, based on the injectivity condition, it is evident that two non-identical collections of $\{\{A_{ij}\}_{e_{ij} \in E}\}$ would yield two different embedded feature vectors.

Furthermore, in the first part, we have proven the existence of at least two distinct local 3D subgraphs with isometric local tree structures such that the corresponding geometric weights $\{\{A_{ij}\}_{e_{ij} \in E}\}$ generated by the encoder $\phi$ are different. Combining both parts, we have successfully demonstrated the theorem.

$\square$

# E  Related proofs and discussions of Section 4

**Torsion Angle is Secretly Hidden in FT**

Recall the edge-wise (signed) torsion angle $\tau_{ij}$ [58] involves the 1-hop atom pairs $i$ and $j$ and two 2-hop atoms $k$ and $l$, then $\tau_{ij}$ is defined to be the dihedral angle between plane $k - i - j$ and plane $l - j - i$. Although exhausting all torsion angles requires $O(k^2)$ complexity, Wang et al. [23] reduces the computation to $O(k)$ order by selecting a canonical 2-hop atom $k$ and $l$, which is enough for detecting the relative orientations between atoms (insufficient for general tasks like many body interactions).

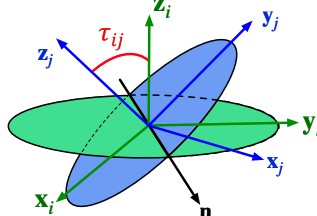

Figure 5: $\tau_{ij}$ indicates the relative rotation of two frames along the z-axis.

Now we show how $\tau_{ij}$ naturally appears as one of the derivatives from frame transition functions. For node $i$, define the equivariant frame $\mathcal{F}_i$ by

$$(\mathbf{e}_1^i, \mathbf{e}_2^i, \mathbf{e}_3^i) = (\mathbf{x}_i - \mathbf{x}_j, \mathbf{x}_i - \mathbf{x}_k, \mathbf{e}_1^i \times \mathbf{e}_2^i).$$

$\mathcal{F}_i$ is normalized through the Gram-Schmidt algorithm. For node j, $\mathcal{F}_j$ is defined similarly by

$$(\mathbf{e}_1^j, \mathbf{e}_2^j, \mathbf{e}_3^j) = (\mathbf{x}_j - \mathbf{x}_i, \mathbf{x}_j - \mathbf{x}_l, \mathbf{e}_1^j \times \mathbf{e}_2^j).$$

Then following the transition formula (5),

$$R_{ij} = (\mathbf{e}_1^i, \mathbf{e}_2^i, \mathbf{e}_3^i) \cdot (\mathbf{e}_1^j, \mathbf{e}_2^j, \mathbf{e}_3^j)^T.$$

Note that for an orthonormal matrix, its inverse is equal to its transpose. Therefore, based on the standard definition of a dihedral angle, we have:

$$\tau_{ij} = \mathbf{e}_3^i \cdot \mathbf{e}_3^j \equiv R_{ij}(3, 3).$$

In conclusion, $\tau_{ij}$ represents only one component of the transition matrix $R_{ij}$. To fully determine $R_{ij}$, we still require two additional angles, as a transition matrix is uniquely determined by three Euler angles.

**Proof of Theorem 4.1.**

**Remark.**

1. The crucial property we leverage to prove this theorem is that all local scalars can be expressed through local scalarization using one local frame and a nonlinear transformation. However, it is important to note that the transition between local frames reveals the relative change of local geometries, which is inherently non-local.

2. Interestingly, stemming from the fact that invariant scalars can be expressed using scalarization and a universal approximator, we can deduce that by incorporating the frame transition matrix into the union set of $S_{\mathbf{B}}$ and $S_{\mathbf{C}}$, all (continuous) invariant interactions $f_a(\mathbf{B}, \mathbf{C})$ can be expressed.

*Proof.* This theorem is proved in two steps:

1. The first step characterizes all scalars $S_{\mathbf{B}}$ ($S_{\mathbf{C}}$) determined by isolated local clusters $\mathbf{B}$ and $\mathbf{C}$ through equivariant frames and local scalarization;

2. The second step constructs a specific invariant function $f_a(\mathbf{B}, \mathbf{C})$ that cannot be expressed by taking the union of $S_{\mathbf{B}}$ and $S_{\mathbf{C}}$: $\{S_{\mathbf{B}} \parallel S_{\mathbf{C}}\}$.

Let $G$ denote a 3D point cloud. As proven in Du et al. [30], when equipped with an equivariant frame $\mathbf{F}_G$, all equivariant features of $G$ can be transformed into scalar features through scalarization without information loss. More precisely, follow the convention in the main text, let $\tilde{G}$ be the output of performing scalarization (using any equivariant frames purely determined by $G$, see Section B for examples) on the 3D point cloud $G$. For any invariant function $f(G)$, there exists a corresponding function $\tilde{f}$ such that:

$$f(G) = \tilde{f}(\tilde{G}). \tag{17}$$

Since $S_G$ represents the collection of all invariant scalars produced by $G$, which is equivalent to the collection of all invariant functions that depend on the 3D point cloud $G$, Eq. 17 implies that $S_G$ can be generated by a finite set of invariant scalars $\tilde{G}$. To apply this insight to our current theorem, we have two different 3D clouds $\mathbf{B}$ and $\mathbf{C}$. Therefore, we need to build two local equivariant frames $\mathcal{F}_{\mathbf{B}} = (e_1^{\mathbf{B}}, e_2^{\mathbf{B}}, e_3^{\mathbf{B}})$ and $\mathcal{F}_{\mathbf{C}} = (e_1^{\mathbf{C}}, e_2^{\mathbf{C}}, e_3^{\mathbf{C}})$. Importantly, the frame $\mathcal{F}_{\mathbf{B}}$ itself doesn't depend on $\mathbf{C}$, and scalarization through $\mathcal{F}_{\mathbf{B}}$ is only performed on the local 3D point cloud $\mathbf{B}$. Therefore, performing operations like scalarizing equivariant information of $\mathbf{C}$ through $\mathcal{F}_{\mathbf{B}}$ would violate the assumptions of the theorem.

We are now ready to construct an explicit counterexample $f_a(\mathbf{B}, \mathbf{C})$ using $\mathcal{F}_{\mathbf{B}}$ and $\mathcal{F}_{\mathbf{C}}$:

$$f_a(\mathbf{B}, \mathbf{C}) := e_1^{\mathbf{B}} \cdot e_1^{\mathbf{C}}.$$

Since $f_a$ is the inner product of two equivariant vectors, it automatically becomes an invariant function. Next, we check whether $f_a(\mathbf{B}, \mathbf{C})$ can be expressed as a function of $f_a(\{S_{\mathbf{B}} \parallel S_{\mathbf{C}}\})$. The equivariant component of $f_a$ related to $\mathbf{B}$ is precisely $e_1^{\mathbf{B}}$. Applying the above local scalarization principle, we scalarize $e_1^{\mathbf{B}}$ through $\mathcal{F}_{\mathbf{B}}$ and obtain:

$$e_1^{\mathbf{B}} \rightarrow \tilde{e}_1^{B} = (1, 0, 0).$$

Similarly, $e_1^{\mathbf{C}}$ is also transformed into a constant scalar tuple $\tilde{e}_1^{\mathbf{C}} = (1, 0, 0)$ through $\mathcal{F}_{\mathbf{C}}$. As constant inputs generate constant outputs, we conclude that the derived local scalars can only approximate constant functions. However, since the local frames change as we vary the 3D structure of $\mathbf{B}$ and $\mathbf{C}$, it is evident that $e_1^{\mathbf{B}} \cdot e_1^{\mathbf{C}}$ is not a constant function of $(\mathbf{B}, \mathbf{C})$. Thus, we complete the proof by contradiction.

Similarly, $e_1^{\mathbf{C}}$ is also transformed to a constant scalar tuple $\tilde{e}_1^{\mathbf{C}} = (1, 0, 0)$ through $\mathcal{F}_{\mathbf{C}}$. As constant inputs generate constant outputs, we conclude that the deduced local scalars can only approximate constant functions. However, since the local frames are changing as we vary the 3D structure of $\mathbf{B}$ and $\mathbf{C}$, it's obvious that $e_1^{\mathbf{B}} \cdot e_1^{\mathbf{C}}$ is not a constant function of $(\mathbf{B}, \mathbf{C})$. Therefore, we finish the proof by contradiction. $\square$

**Further Comments.**

1. It is important to note that the counterexample we constructed is not the only valid counterexample. For instance, the norm of any tensors that depend on both **B** and **C** (e.g., $\left\| e_1^{\mathbf{B}} + e_1^{\mathbf{C}} \right\|$, $\left\| e_1^{\mathbf{B}} \otimes e_2^{\mathbf{B}} \otimes e_2^{\mathbf{C}} \otimes e_3^{\mathbf{C}} \right\|$) can also serve as valid counterexamples.

2. Let $R_{\mathbf{BC}}$ denote the frame transition function between $\mathcal{F}_{\mathbf{B}}$ and $\mathcal{F}_{\mathbf{C}}$, defined in the same way as in Equation 5. The proof of Theorem 4.1 implies that **all types of invariant interactions $f_a(\mathbf{B}, \mathbf{C})$ can be expressed solely as functions of the union of three sets $S_{\mathbf{B}}$, $S_{\mathbf{C}}$, and $R_{\mathbf{BC}}$** : $\{ S_{\mathbf{B}} \parallel S_{\mathbf{C}} \parallel R_{\mathbf{BC}} \}$. This is because, through $R_{\mathbf{BC}}$, the 3D point cloud **C** can be scalarized by the local frame $\mathcal{F}_{\mathbf{B}}$. Specifically, let $\tilde{S}_{\mathbf{C}}^B$, $\tilde{S}_{\mathbf{C}}^C$ denote the scalarized 3D point cloud of $S_{\mathbf{C}}$ with respect to $\mathcal{F}_{\mathbf{B}}$ and $\mathcal{F}_{\mathbf{C}}$, respectively. We have the following diagram:

$$S_{\mathbf{C}} \xrightarrow{\mathcal{F}_{\mathbf{C}}} \tilde{S}_{\mathbf{C}}^C \xrightarrow{R_{\mathbf{BC}}} \tilde{S}_{\mathbf{C}}^B.$$

Utilizing formula 15 on the union of the two 3D point clouds, $(\mathbf{B}, \mathbf{C})$, leads us to this conclusion.

**Realizing FT by Equivariant Messages:** From the **FT** definition 5, each element of the $3 \times 3$ matrix $R_{ij}$ is calculated by

$$R_{ij}(k, l) = \mathbf{e}_k^i \cdot \mathbf{e}_l^j. \tag{18}$$

Now we show how to reproduce $R_{ij}(k, l)$ through equivariant messages. Let the equivariant message $\mathbf{m}_i$ be the following:

$$\mathbf{m}_i := (\mathbf{e}_1^i, \mathbf{e}_2^i, \mathbf{e}_3^i) \cdot \begin{bmatrix} 1 & 1 & 1 & 0 & 0 & 0 & 0 & 0 & 0 \\ 0 & 0 & 0 & 1 & 1 & 1 & 0 & 0 & 0 \\ 0 & 0 & 0 & 0 & 0 & 0 & 1 & 1 & 1 \end{bmatrix}$$

It's easy to check that $\mathbf{m}_i \in \mathbf{R}^{3 \times 9}$ consists of 9 equivariant vectors (**multi-channels**). For atom $j$, $\mathbf{m}_j$ is defined symmetrically. For each node, we also store the scalar messages, e.g., $\left\| \mathbf{e}_k^i \right\|$ for $1 \leq k \leq 3$. Flattening the whole matrix $R_{ij}$ into a $\mathbf{R}^{1 \times 9}$ array, then $R_{ij}$ is obtained by simple summation and taking the vector norm:

$$\|\mathbf{m}_i + \mathbf{m}_j\| = \left\{ \left\| \mathbf{e}_k^i + \mathbf{e}_l^j \right\| \right\}_{1 \leq k, l \leq 3},$$

where the norm is taken for each column of $\mathbf{m}_i + \mathbf{m}_j$, such that $\|\mathbf{m}_i + \mathbf{m}_j\| \in \mathbf{R}^{1 \times 9}$. Then,

$$R_{ij} = \frac{1}{2} \left[ \left\| \mathbf{e}_k^i + \mathbf{e}_l^j \right\|^2 - \left\| \mathbf{e}_k^i \right\|^2 - \left\| \mathbf{e}_l^j \right\|^2 \right].$$

Our illustration also demonstrates the importance of keeping multi-channel tensor messages.

**Relation with Previous Equivariant Update Methods.** Following the efficiency principle established in section 4, we don't encode the data of the transition matrices explicitly. Instead, we implement tensor messages to fill in the expressiveness gap. Among the tremendously different designs of equivariant graph neural networks, Schütt et al. [35] is closely related to our equivariant updating method. By the above argument, the inner product operation for node $i$ (see (9) of Schütt et al. [35])

$$< \mathbf{U}\mathbf{v}_i, \mathbf{V}\mathbf{v}_i >$$

can also be reinterpreted as a realization of the (aggregated) frame transition matrix (5).

Moreover, since the equivariant vectors $\mathbf{U}\mathbf{v}_i$ and $\mathbf{V}\mathbf{v}_i$ are both aggregated vector features that belong to the same node $i$ and the inner product operation between them is performed in the node-wise updating phase, Schütt et al. [35] actually avoids the 2-hop $O(k^2)$ complexity of computing $R_{xy}$ for all neighborhood node pairs $(\mathbf{x}, \mathbf{y})$ (while able to express the torsion angle implicitly). For our algorithm, we utilize the scalarization and tensorization in the node-wise updating phase. By the universal approximation theorem 5.1, our method can approximate any inner product operations.

## F   Related proofs and discussions of Section 5

**Equivariant Frames and Higher Order Scalarization and Tensorization.** Given an edge $e_{ij}$ with two atom positions $(\mathbf{x}_i, \mathbf{x}_j)$, we define our edge-wise $SE(3)$ equivariant frames $\mathcal{F}_{ij}$ as follows:

$$(\mathbf{e}_1, \mathbf{e}_2, \mathbf{e}_3) = (\frac{\mathbf{x}_i - \mathbf{x}_j}{\|\mathbf{x}_i - \mathbf{x}_j\|}, \frac{\mathbf{x}_i \times \mathbf{x}_j}{\|\mathbf{x}_i \times \mathbf{x}_j\|}, \frac{\mathbf{x}_i - \mathbf{x}_j}{\|\mathbf{x}_i - \mathbf{x}_j\|} \times \frac{\mathbf{x}_i \times \mathbf{x}_j}{\|\mathbf{x}_i \times \mathbf{x}_j\|}). \tag{19}$$

To ensure frame translation invariance, we adopt the approach followed by previous works [59, 60] by restricting the entire 3D conformer space to a linear subspace where the center of mass (CoM) of the system (either the entire system or the sub-cluster to which $i$ and $j$ belong) is set to zero. Alternatively, constructing an $E(3)$ frame is also possible but requires an additional atom position $\mathbf{x}_k$, which can be selected using the K-Nearest Neighbor algorithm. If $(\mathbf{x}_i, \mathbf{x}_j, \mathbf{x}_k)$ spans the 3D space, we obtain an $E(3)$ equivariant frame by performing Gram-Schmidt orthogonalization. For different constructions of $E(3)$ frames, readers can refer to Wang and Zhang [61]. However, in this work, we focus on $SE(3)$ frames since molecular 3D conformers exhibit $SE(3)$ symmetry but do not possess reflection symmetry.

Once we have an equivariant frame, every vector is a linear combination of the three orthogonal vectors in the frame. Moreover, the unique combination coefficients are exactly the 'scalarized' coordinates in (8). A similar procedure also applies to higher order tensors. Indeed, the vector frame $\mathcal{F}^1$ extends to a tensor frame $\mathcal{F}^r$ of arbitrary order $r > 1$:

$$\mathcal{F}^r := \{\mathbf{e}_{1_1} \otimes \cdots \otimes \mathbf{e}_{1_r}\}_{1 \leq i_1, \ldots, i_r \leq 3}. \tag{20}$$

Since the orthonormal frame $\mathcal{F}^r$ is complete in the sense that it spans the whole tensor space of order $r$, every r-th order tensor admits a unique decomposition:

$$\mathbf{T} = \sum_{1 \leq i_1, \ldots, i_r \leq 3} T^{i_1, \ldots, i_r} \mathbf{e}_{i_1} \otimes \cdots \otimes \mathbf{e}_{i_r}. \tag{21}$$

It's easy to prove that the collection $\{T^{i_1, \ldots, i_r}\}_{1 \leq i_1, \ldots, i_r \leq 3}$ consists of invariant scalars. We call the process from $\mathbf{T}$ to $\{T^{i_1, \ldots, i_r}\}_{1 \leq i_1, \ldots, i_r \leq 3}$ **scalaraization**.

**Tensorization** is the inverse of scalarization, in the sense that it sends scalars $\{T^{i_1, \ldots, i_r}\}_{1 \leq i_1, \ldots, i_r \leq 3}$ to tensor $\mathbf{T}$. Under the same frames we use during scalarization, the following diagram demonstrates the pipeline of producing $L$ second-order tensors out of $\{T_j^{i_1 i_2}\}_{1 \leq i_1, i_2 \leq 3}$:

$$\{\mathbf{T}_1, \ldots, \mathbf{T}_L\} = \underbrace{\left\{ \begin{bmatrix} T_1^{11}, & T_1^{12}, & T_1^{13} \\ T_1^{21}, & T_1^{22}, & T_1^{23} \\ T_1^{31}, & T_1^{32}, & T_1^{33} \end{bmatrix}, \ldots, \begin{bmatrix} T_L^{11}, & T_L^{12}, & T_L^{13} \\ T_L^{21}, & T_L^{22}, & T_L^{23} \\ T_L^{31}, & T_L^{32}, & T_L^{33} \end{bmatrix} \right\}}_{L \text{ channels}} \odot \begin{bmatrix} \mathbf{e}_1 \otimes \mathbf{e}_1, & \mathbf{e}_1 \otimes \mathbf{e}_2, & \mathbf{e}_1 \otimes \mathbf{e}_3 \\ \mathbf{e}_2 \otimes \mathbf{e}_1, & \mathbf{e}_2 \otimes \mathbf{e}_2, & \mathbf{e}_2 \otimes \mathbf{e}_3 \\ \mathbf{e}_3 \otimes \mathbf{e}_1, & \mathbf{e}_3 \otimes \mathbf{e}_2, & \mathbf{e}_3 \otimes \mathbf{e}_3 \end{bmatrix},$$
$$\tag{22}$$

where $\odot$ denotes the element-wise product.

**Proof of Theorem 5.1**

*Proof.* The proof relies on the invertibility of **Scalarization** and **Tensorization** operations, as demonstrated in Appendix A.5 of Du et al. [30]. This invertibility allows us to establish a commutative diagram as follows:

$$\begin{array}{ccc} \mathbf{T}^{l-1} & \xrightarrow{\rho} & \mathbf{T}^l \\ \downarrow \text{\scriptsize Scalarize} & & \uparrow \text{\scriptsize Tensorize} \\ \tilde{T}^{l-1} & \xrightarrow{\text{\scriptsize MLP}} & \tilde{T}^{l-1}. \end{array}$$

This diagram illustrates that for each mapping $\rho$, there exists a corresponding "scalarized" mapping $\tilde{\rho}$ given by:

$$\tilde{\rho} := \textbf{Tensorize} \circ \rho \circ \textbf{Scalarize}.$$

By applying **Tensorize**, followed by $\rho$, and then **Scalarize**, we obtain an invariant representation $\tilde{\rho}$. Since **MLP** serves as a universal approximator of invariant functions, we can always find an **MLP** that expresses $\tilde{\rho}$. By reversing the arrows, we conclude the proof. □

**Proof of Equivariance for LEFTNet** LEFTNet consists of multiple layers of **LSE** and **FTE**. **LSE** is realized by scalarization, and **FTE** is realized by scalarization and tensorization. Since the invariance of scalarization and the equivariance of tensorization have been proved, we finish the proof.

# G Extended Related Work

Table 4: Categorization of representative geometric GNN algorithms. $^*$ denotes partially satisfying the requirement.

| Method | Symmetry | LSE | FTE | Complexity |
|---|---|---|---|---|
| SchNet [28] | E(3)-invariant | ✗ | ✗ | $O(nk)$ |
| EGNN [14] | E(3)-equivariant | ✗ | ✓$^*$ | $O(nk)$ |
| GVP-GNN [31] | E(3)-equivariant | ✗ | ✓ | $O(nk)$ |
| ClofNet [30] | SE(3)-equivariant | ✗ | ✗ | $O(nk)$ |
| PaiNN [35] | E(3)-equivariant | ✗ | ✓ | $O(nk)$ |
| ComENet [23] | SE(3)-invariant | ✓ | ✓$^*$ | $O(nk)$ |
| TFN [13] | SE(3)/E(3)-equivariant | ✗ | ✓ | $O(nk)$ |
| Equiformer [36] | SE(3)/E(3)-equivariant | ✗ | ✓ | $O(nk)$ |
| SphereNet [15] | SE(3)-invariant | ✓$^*$ | ✓$^*$ | $O(nk^2)$ |
| GemNet [37] | SE(3)-invariant | ✓$^*$ | ✓$^*$ | $O(nk^3)$ |
| LEFTNet (Ours) | SE(3)/E(3)-equivariant | ✓ | ✓ | $O(nk)$ |

Based on the discussions in Section 3 and 4, we identify two essential components for constructing expressive equivariant 3D Graph Neural Networks (GNNs): (1) Local 3D Substructure Encodings (**LSE**), which enable the local message passing to capture diverse local 3D structures; and (2) Frame Transition Encodings (**FTE**), which incorporate equivariant coordinate transformations between different local patches into the 3D GNN.

## G.1 Modular overview of 3D GNN

Based on the general notion of Local 3D Substructure Encodings (**LSE**) and Frame Transition Encodings (**FTE**), we have provided concrete constructions to realize them. However, it is important to note that there are also implicit methods to encode the information of **LSE** and **FTE**. In order to provide a comprehensive overview, we review previous 3D Graph Neural Networks (GNNs) that follow this framework and summarize the findings in Table 4. To ensure a fair comparison, we include the computational complexity as it is often a trade-off with expressiveness. A detailed analysis of this trade-off is provided at the end of Section G.2.

Regarding Local 3D Substructure Encodings (**LSE**), SphereNet [15] and GemNet [37] (implicitly) encode local 3D substructures through a computation-intensive edge-based update. We indicate these two architectures with a $*$ in Table 4 due to the following reasons: In the case of GemNet [37], the message passing framework considers the 1-hop neighborhood with respect to both end points of an edge $e_{ij}$, which inherently contains all the nodes of the mutual 3D subgraph $S_{i-j}$. However, when compared to SphereNet [15], which encodes all node positions under an angular coordinate system, GemNet [37] only captures partial geometric information. On the other hand, SphereNet [15] utilizes an edge-based update that aggregates 1-hop neighbors with respect to the source node. While this approach considers the immediate neighbors, it may not include all the nodes inside the subgraph $S_{i-j}$ (e.g., the neibors of the target node).

Regarding Frame Transition Encodings (**FTE**), we have demonstrated (see the proof of Theorem 4.1) that the essential information of **FTE** does not rely on the specific choices of local **equivariant** frames. Therefore, most 3D GNNs with equivariant vector updates that can incorporate at least three (**multi-channel**) independent equivariant vectors as edge features are capable of expressing local frame transitions (**FT**). However, EGNN [14] is an exception as it only updates the position vector (i.e., one channel), which is insufficient for capturing the entire **FT**. In other words, the power of the update function $\phi$ in Equation (6) also influences the encoding of **FT**. Additionally, models that encode torsion angle information partially express **FTE**, as illustrated in Appendix E. While there are multiple ways to realize **LSE** and **FTE**, there is a trade-off between efficiency and expressiveness in

terms of the number of hops considered for message passing, as indicated in the last column of Table 4.

In contrast to our invariant realization of **LSE**, Batatia et al. [24] constructs their framework by building a complete $(E(3)$ + permutation) equivariant polynomial basis using spherical harmonics and tensor product, where the monomial variables are a combination of 3D features from different nodes (bodies). On the other hand, we achieve the function of **LSE** and **FTE** through edgewise scalarization $A_{ij}$ and equivariant message passing (see Fig. 3). While Batatia et al. [24] utilize the atomic cluster expansion (ACE) mechanism, we provide an illustration of how to equivariantly realize **LSE** based on local ACE in the proof of Theorem 3.1. Another work Wang and Zhang [61] uses an ensemble of frames to model the local environment defined through a distance cutoff, which is suitable for pure 3D geometry. However, our local 3D hierarchy incorporates well with the 2D topology. Therefore, whether [61] encodes **LSE** depends on the radius of the local environment and whether the ensemble of 'frames' purely built as a combination of radical directions (see Equation (6) in [61]) spans $\mathbf{R}^3$. In comparison to geometric GNNs like SphereNet [15] and GemNet [37], which encode reflection-antisymmetric torsion angles, and LeftNet, which implements $SE(3)$ equivariant, reflection-antisymmetric frames, the ensemble of frames in [61] is reflection symmetric and therefore cannot differentiate local chemical isomers. Additionally, Wang and Zhang [61] defines the '(ensembled) frame to frame' projection, which is somehow equivalent to our **FTE**, and utilizes it to detect the global isomorphism from local observations. In contrast, our **FTE** is employed to bridge the expressiveness gap of global continuous functions, such as regression tasks on graphs (see the comments in Section E), rather than focusing on detecting global isomorphisms, as in classification tasks on graphs. Moreover, we have demonstrated that using one $SE(3)$ equivariant frame for each local patch is sufficient for achieving local expressiveness, considering efficiency as well. Finally, we provide another invariant realization of **FTE**, inspired by [62], which is presented at the end of Section H.

### G.2 Relationship to Geometric WL test

Recently, Joshi et al. [22] propose a **geometric k-WL test** (GWL) to assess the expressiveness of geometric GNN algorithms. In essence, our tree isomorphism aligns with the 1-hop geometric isomorphism introduced in GWL, while the fine-grained triangular isomorphism falls between the 1-hop and 2-hop geometric isomorphism described in GWL. From a model design perspective, we achieve the realization of **LSE** through local scalarization, which guarantees expressiveness through the Kolmogorov representation theorem ( Zaheer et al. [57]) and the universal approximator property of MLP. Different from our geometry perspective, the fundamental concepts of body order and tensor order, which stem from classical inter-atomic potential theories and exhibit equivariance, play a crucial role in measuring expressive power in Joshi et al. [22]. In addition to the local geometric isomorphism hierarchy, we discover the significance of **FTE** as the connecting bridge between local invariant scalars and global geometric expressiveness. This realization, along with **LSE** on mutual 3D substructures, sheds light on the insufficiency of the 1-hop local scalarization implemented in ClofNet ( Du et al. [30]). We further explore the connection between **FTE** and the neural sheaf Laplacian in Section H.

**Computational efficiency of LEFTNet**  In the analysis of a graph with $n$ nodes and an average of $k$ edges per node, the computational efficiency of a message passing-based graph neural network (MPNN) is typically described by the following form:

$$\mathcal{O}(fn + gn \cdot k^c), \tag{23}$$

where $f$ represents the computational cost of the node-level updating function (usually an MLP), and $g$ represents the computational cost of calculating the message for each edge. The power $c$ is a crucial factor that determines the computational cost of performing each message passing operation. For instance, SphereNet (Liu et al., 2021) aggregates all 1-hop nodes with respect to the source node of an edge to construct a message, resulting in a complexity of $\mathcal{O}(nk^2)$ for message passing.

To analyze the computational complexity of LEFTNet, we can divide it into the following computational steps:

1. **Building the $S_{i-j}$ and scalarizing with respect to edge frame $\mathbf{F}_{ij}$.**  The 3D subgraph $S_{i-j}$ is constructed by intersecting the 1-hop neighbors $\mathcal{N}_i$ and $\mathcal{N}_j$, which requires $\mathcal{O}(nk)$

computations. Additionally, collecting the 1-hop neighbors for each node has a complexity of $\mathcal{O}(nk)$. Finally, scalarizing each $S_{i-j}$ with the edge-wise frame $\mathbf{F}_{ij}$ (which involves taking inner products) and extending the result to $m$ channels has a complexity of $\mathcal{O}(mnk)$. Therefore, the total complexity for this step remains at $\mathcal{O}(mnk)$.

2. **Equivariant message passing.** From the first step, we get the scalarized $\tilde{S}_{i-j}$ from the 3D subgraph $S_{i-j}$. Then, the edge-wise message is obtained by transforming the scalarized $\tilde{S}_{i-j}$ with a **MLP**, and the output invariant scalars is multiplied with both the invariant and equivariant node features (let $d$ denotes the dimension of the node feature, then the multiplication's complexity is $\mathcal{O}(dnk)$. Let $l$ denotes the complexity of **MLP** (depends on the depth and hidden dimension of **MLP**), then the message passing takes $\mathcal{O}((d+l)nk)$ computation.

3. **Node level updating.** This step involves nodewise scalarization, applying an **MLP**, and performing tensorization. Suppose the input and output channel numbers for equivariant tensor features are both $d$, then the scalarization using the node frame (similar to step 2) has a complexity of $\mathcal{O}(dn)$ computations. Similarly, tensorization, which is the inverse operation of scalarization, also has a complexity of $\mathcal{O}(dn)$. Therefore, the total complexity for this step is $\mathcal{O}((l+d)n)$.

In conclusion, we have analyzed the three steps required for performing one layer of LEFTNet. It is worth noting that the efficiency of LEFTNet is comparable to other 1-hop based GNN algorithms like SchNet [28] ($c = 1$ in Eq. 23). The final complexity is $\mathcal{O}(ldn + (d + m + l)nk)$, which is summarized in Table 4.

# H  Neural sheaf interpretation

Since our invariant scalarization, utilized in the LSE module, shares similarities with the scalarization technique on vector sheaves (see Hsu [52]), it is natural to explore a potential interpretation of LEFTNET within the framework of neural sheaf diffusion proposed by Bodnar et al. [62]. In this section, we aim to modify the neural sheaf diffusion architecture to preserve $SE(3)$ equivariance by incorporating our node and edge frames.

We begin by revisiting the concept of a neural sheaf and its associated sheaf Laplacian operator. A *cellular sheaf* over a discrete graph is a mathematical object that assigns a vector space to each node and edge in the graph and specifies a linear map between these spaces for each incident node-edge pair:

**Definition H.1.** *A cellular sheaf* $(G, \mathcal{F})$ *on a graph* $G = (V, E)$ *consists of:*

- *A vector space* $\mathcal{F}(v)$ *for each* $v \in V$.

- *A vector space* $\mathcal{F}(e)$ *for each* $e \in E$.

- *A linear map* $\mathcal{F}_{v \trianglelefteq e} : \mathcal{F}(v) \to \mathcal{F}(e)$ *for each incident* $v \trianglelefteq e$ *node-edge pair.*

In the case of a 3D graph neural network (GNN), the node-wise features typically consist of tensors (as defined in 10), which naturally form vector spaces. The key aspect of defining a sheaf lies in determining the specific assignments of the linear maps $\mathcal{F}_{v \trianglelefteq e}$ for each node $v$ and each edge $e$. For non-geometric GNNs, $\mathcal{F}_{v \trianglelefteq e}$ captures how the opinions (represented as vectors in $\mathcal{F}(v)$) manifest in a "discourse space" formed by $\mathcal{F}(e)$. In our approach, we equip each node with an equivariant frame denoted by $F_v$ and assign an equivariant frame to each edge denoted by $F_e$. By definition, $F_v$ and $F_e$ capture the local geometry around the node and the edge, respectively. Thus, a natural choice for $\mathcal{F}_{v \trianglelefteq e}$ is given by:

$$\mathcal{F}_{v \trianglelefteq e} := F_v^T F_e \in \mathcal{O}(3).$$

To ensure $SE(3)$ equivariance in the sheaf structure, we define $\mathcal{F}(v)$ as the set of scalarized vectors obtained by applying scalarization using $F_v$. It is important to note that the linear map $\mathcal{F}_{v \trianglelefteq e}$ between vectors can be extended to a linear map between higher-order tensors through tensor product operations (see Section F). This construction guarantees the entire structure to be $SE(3)$-invariant. With a well-defined sheaf structure, we can introduce the sheaf Laplacian operator that defines a

transformation between node features:

$$L_{\mathcal{F}}(\boldsymbol{x})_v := \sum_{v,u \trianglelefteq e} \mathcal{F}_{v \trianglelefteq e}^{\top}(\mathcal{F}_{v \trianglelefteq e}\boldsymbol{x}_v - \mathcal{F}_{u \trianglelefteq e}\boldsymbol{x}_u).$$

In our case, the sheaf Laplacian can be expressed as:

$$L_{\mathcal{F}}(\boldsymbol{x})_v = \sum_{v,u \trianglelefteq e} \boldsymbol{x}_v - F_v^T F_u \boldsymbol{x}_u \ , \tag{24}$$

using the properties of orthogonal transformations. Thus, this operator quantifies the collective "disagreement of local geometries" at each node. Neural sheaf diffusion propagates information through nodes using the following partial differential equation (PDE):

$$\boldsymbol{X}(0) = \boldsymbol{X}, \quad \dot{\boldsymbol{X}}(t) = -L_{\mathcal{F}}\boldsymbol{X}(t). \tag{25}$$

Here, $\boldsymbol{X}$ represents the node feature matrix. In accordance with the nonlinear parametrization proposed in [62], we have the following model:

$$\boldsymbol{X}(t+1) = \sigma\Big(\big(\boldsymbol{I}_{nd} - L_{\mathcal{F}}\big)(\boldsymbol{I}_n \otimes \boldsymbol{W}_1)\boldsymbol{X}(t)\boldsymbol{W}_2\Big). \tag{26}$$

where any applicable nonlinear activation function $\sigma$ can be utilized since our cellular sheaf and its Laplacian are invariant. It is important to note that the key component of $L_{\mathcal{F}}$ (denoted by $F_v^T F_u$ in Wq. 24) corresponds to the frame transition (**FTE**) module between node frames. Hence, Equation (26) provides an invariant realization of the **FTE** module based on equivariant node frames. For equivariant outputs, tensorization can be applied after the final layer of Equation (26). To incorporate the **LSE** module, we can include the **LSE** information as a part of invariant node features to $\mathcal{F}_v$ prior to the message aggregation step in Equation (26).

# I   Additional experiments

**Experiment Detail for Table 1 and 2.** For QM9, baseline results are taken from Liao and Smidt [36]. For MD17, baseline results are taken from the original papers (with unit conversions if needed). All models are trained on energies and forces, and WoFE is the weight of force over energy in loss functions. All experiments are conducted on a single NVIDIA GeForce RTX 2080 Ti 11GB GPU. Our implementation is based on the libraries including PyTorch [63], PyG [64], and DIG [65].

**Ablation Study.** As discussed in Section 5, there are two main modules in LEFTNet, namely **LSE** and **FTE**. We conduct experiments on QM9 and MD17 to show the importance of each component. Experimental results are summarized in Table 5 and Table 6. The results show that using **LSE** can outperform the model without both **LSE** and **FTE** on all tasks. Adding **FTE** can further improve the performance. The results demonstrate the importance of **LSE** and **FTE** modules.

Table 5: Ablation study on QM9 dataset. The evaluation metric is MAE for each property. The **best** performances are bolded and the second best are underlined. Detailed LEFTNet (**LSE** only) is shown in Algorithm 2. LEFTNet (**LSE** + vector **FTE**) is our LEFTNet introduced in the main paper, and the detailed algorithm is in Algorithm 1. LEFTNet (**LSE** + tensor **FTE**) means the message passing and updating contains higher order tensors, built by Eq. 22.

| Task | $\alpha$ | $\Delta\varepsilon$ | $\varepsilon_{\mathrm{HOMO}}$ | $\varepsilon_{\mathrm{LUMO}}$ | $\mu$ | $C_\nu$ | $G$ | $H$ | $R^2$ | $U$ | $U_0$ | ZPVE |
| Units | bohr$^3$ | meV | meV | meV | D | cal/mol K | meV | meV | bohr$^3$ | meV | meV | meV |
| LEFTNet (w/o **LSE** and **FTE**) | .053 | 49 | 33 | 25 | .038 | .026 | 9 | 8 | .425 | 8 | 8 | 1.59 |
| LEFTNet (**LSE** only) | .043 | 49 | 31 | 23 | .031 | .025 | 8 | 7 | .156 | 8 | 7 | 1.34 |
| LEFTNet (**LSE** + vector **FTE**) | .039 | 39 | 23 | 18 | **.011** | **.022** | **6** | **5** | **.094** | **5** | 5 | **1.19** |
| LEFTNet (**LSE** + tensor **FTE**) | **.038** | 38 | 22 | 17 | **.011** | **.022** | 7 | 6 | .096 | **5** | 6 | 1.20 |

**Results on rMD17.** Following Batatia et al. [24], we conduct experiments on rMD17 to compare with recent studies. Results show that our LEFTNet can achieve comparable performance to state-of-the-art methods such as MACE and NequIP, while outperforming other baseline methods like GemNet and PaiNN.

**Model and training hyperparameters.** Model and training hyperparameters for our method on different datasets are listed in Table 8.

Table 6: Abalation Study on MD17 dataset. The evaluation metric is MAE for per-atom forces prediction (kcal/mol Å). The **best** performances are bolded and the second best are underlined. Detailed LEFTNet (**LSE** only) is shown in Algorithm 2. LEFTNet (**LSE** + vector **FTE**) is our LEFTNet introduced in the main paper, and detailed algorithm is in Algorithm 1. LEFTNet (**LSE** + tensor **FTE**) means the message passing and updating contains higher order tensors, built by Eq. 22.

| Molecule | LEFTNet (w/o **LSE** and **FTE**) | LEFTNet (**LSE** only) | LEFTNet (**LSE** + vector **FTE**) | LEFTNet (**LSE** + tensor **FTE**) |
|---|---|---|---|---|
| Aspirin | 1.083 | 0.451 | 0.300 | **0.210** |
| Benzene | 0.425 | 0.185 | **0.145** | 0.176 |
| Ethanol | 0.341 | 0.149 | 0.138 | **0.118** |
| Malonaldehyde | 0.594 | 0.276 | 0.209 | **0.159** |
| Naphthalene | 0.658 | 0.175 | 0.073 | **0.063** |
| Salicylic acid | 0.828 | 0.313 | 0.167 | **0.141** |
| Toluene | 0.625 | 0.166 | 0.084 | **0.070** |
| Uracil | 0.581 | 0.206 | **0.116** | 0.117 |

Table 7: Mean Absolute Error for energy(meV) per-atom forces prediction (meV Å) on rMD17 dataset. Baseline results are taken from Batatia et al. [24]. The best results are **bolded**.

| | | LEFTNet | MACE | Allegro | BOTNet | NequIP | GemNet (T/Q) | ACE | FCHL | GAP | ANI | PaiNN |
|---|---|---|---|---|---|---|---|---|---|---|---|---|
| Aspirin | E | **2.1** | 2.2 | 2.3 | 2.3 | 2.3 | - | 6.1 | 6.2 | 17.7 | 16.6 | 6.9 |
| | F | **6.4** | 6.6 | 7.3 | 8.5 | 8.2 | 9.5 | 17.9 | 20.9 | 44.9 | 40.6 | 16.1 |
| Azobenzene | E | **0.7** | 1.2 | 1.2 | **0.7** | **0.7** | - | 3.6 | 2.8 | 8.5 | 15.9 | - |
| | F | 3.3 | 3.0 | **2.6** | 3.3 | 2.9 | - | 10.9 | 10.8 | 24.5 | 35.4 | - |
| Benzene | E | 0.05 | 0.4 | 0.3 | **0.03** | 0.04 | - | 0.04 | 0.35 | 0.75 | 3.3 | - |
| | F | 0.3 | 0.3 | **0.2** | 0.3 | 0.3 | 0.5 | 0.5 | 2.6 | 6 | 10 | - |
| Ethanol | E | **0.4** | **0.4** | **0.4** | **0.4** | **0.4** | - | 1.2 | 0.9 | 3.5 | 2.5 | 2.7 |
| | F | 3.6 | **2.1** | **2.1** | 3.2 | 2.8 | 3.6 | 7.3 | 6.2 | 18.1 | 13.4 | 10 |
| Malonaldehyde | E | 0.8 | 0.8 | **0.6** | 0.8 | 0.8 | - | 1.7 | 1.5 | 4.8 | 4.6 | 3.9 |
| | F | 5.4 | 4.1 | **3.6** | 5.8 | 5.1 | 6.6 | 11.1 | 10.3 | 26.4 | 24.5 | 13.8 |
| Naphthalene | E | 0.8 | 0.5 | **0.2** | **0.2** | 0.9 | - | 0.9 | 1.2 | 3.8 | 11.3 | 5.1 |
| | F | 1.9 | 1.6 | **0.9** | 1.8 | 1.3 | 1.9 | 5.1 | 6.5 | 16.5 | 29.2 | 3.6 |
| Paracetamol | E | **1.3** | **1.3** | 1.5 | **1.3** | 1.4 | - | 4 | 2.9 | 8.5 | 11.5 | - |
| | F | **4.7** | 4.8 | 4.9 | 5.8 | 5.9 | - | 12.7 | 12.3 | 28.9 | 30.4 | - |
| Salicylic acid | E | 0.9 | 0.9 | 0.9 | 0.8 | **0.7** | - | 1.8 | 1.8 | 5.6 | 9.2 | 4.9 |
| | F | 4.1 | 3.1 | **2.9** | 4.3 | 4 | 5.3 | 9.3 | 9.5 | 24.7 | 29.7 | 9.1 |
| Toluene | E | **0.3** | 0.5 | 0.4 | **0.3** | **0.3** | - | 1.1 | 1.7 | 4 | 7.7 | 4.2 |
| | F | 2.2 | **1.5** | 1.8 | 1.9 | 1.6 | 2.2 | 6.5 | 8.8 | 17.8 | 24.3 | 4.4 |
| Uracil | E | **0.4** | 0.5 | 0.6 | **0.4** | **0.4** | - | 1.1 | 0.6 | 3 | 5.1 | 4.5 |
| | F | 2.8 | 2.1 | **1.8** | 3.2 | 3.1 | 3.8 | 6.6 | 4.2 | 17.6 | 21.4 | 6.1 |

Table 8: Model and training hyperparameters for our method on different tasks.

| Hyperparameter | Values/Search Space | | |
|---|---|---|---|
| | QM9 | MD17 | rMD17 |
| Number of layers | 4, 5, 6 | 4, 6 | 4, 6 |
| Hidden channels | 128, 192, 256 | 256 | 256 |
| Number of radial basis | 24, 32, 96 | 16, 32, 64 | 16, 32, 64 |
| Cutoff | 5, 6, 6.5, 8 | 6, 8, 10 | 6, 8, 10 |
| Epochs | 800 | 1000 | 1000 |
| Batch size | 32 | 1, 4 | 1, 4 |
| Learning rate | 1e-4, 5e-4 | 5e-4 | 5e-4 |
| Learning rate scheduler | steplr | steplr | steplr |
| Learning rate decay factor | 0.5 | 0.5 | 0.5 |
| Learning rate decay epochs | 100 | 200 | 200 |

