# OpenReview forum: "A new perspective on building efficient and expressive 3D equivariant graph neural networks"
_NeurIPS.cc/2023/Conference — NeurIPS 2023 poster_

### Official Review · Reviewer_b3yr · 2023-06-08

**Soundness:** 3 good
**Presentation:** 3 good
**Contribution:** 4 excellent
**Rating:** 8
**Confidence:** 3

**Summary:**

This paper presents the expressive power of equivariant graph neural networks in the context of 3D local isomorphism from a 2D local isomorphism perspective. Similar to the subgraph isomorphism of GNNs, the paper proposes three types of isomorphism in 3D space: Tree Isometric, Triangular Isometric, Subgraph Isometric. Based on the above definitions, the paper defines geometric Weisfeiler-Lehman (WL) tests. Based on the conclusions in paper [1], this paper enhances the expressive power of equivariant GNN by incorporating mutual 3D substructures. On the other hand, to further enhance the expressive power of the equivariant graph neural network, this paper introduces an SE(3)-invariant encoder to exploit mutual 3D structures. Through a theoretical analysis (theoreom 4.1), the paper demonstrates that simple aggregation of local messages cannot approximate global interactions, thus the authors further enhance the model's expressive power through the frame transition matrix. The effectiveness of the proposed model is demonstrated by experimental results on scalar and vector properties of molecules on QM9.

[1] A New Perspective on "How Graph Neural Networks Go Beyond Weisfeiler-Lehman?"

**Strengths:**

1. To the best of my knowledge, this is the first paper to analyze the expressive power of equivariant neural networks from the 3D isomorphism perspective. This could inspire future work to analyze the expressive power of current equivariant neural networks from this perspective.

2. Based on the analysis proposed in the paper on the expressivity of equivariant neural networks, the paper incorporates mutual 3D substructures into 3D GNNs (LSE in this paper) to further enhance their expressive power. The effectiveness of the proposed analysis and methods has been demonstrated experimentally.

3. To reduce the computational cost of neural sheaf diffusion, the paper proposes a novel frame transition matrix to express global geometric information. The computational complexity analysis in line 304 validates the effectiveness of the proposed method.

**Weaknesses:**

1. I have to say that the notation in this paper is confusing, which imposes a burden on me. For instance, h is defined as a node feature on line 81, and f is a tensor-valued function on line 70. However, f is then used as a bijection on line 112. Does this mean the definition of h on line 112 is different from that on line 81?

2. Although the analysis of the expressivity of 3D GNNs is the first from the 3D isomorphism perspective, this paper seems to be based on the analysis in [1]. Moreover, the method proposed by the analysis in [1] can incorporate structural properties into many GNNs, such as GCN, GIN, and GAT. Can this paper similarly incorporate mutual 3D substructures into other 3D GNNs?

3. Recently, some work [2] has analyzed the expressive power of GNNs from the perspective of the attention mechanism, so how can we analyze the expressive power of 3D GNNs from the attention mechanism?



[1] A New Perspective on "How Graph Neural Networks Go Beyond Weisfeiler-Lehman?"

[2] Interpretable and Generalizable Graph Learning via Stochastic Attention Mechanism

**Questions:**

See my comments on weakness.

**Limitations:**

In my personal opinion, one of the limitations of this paper is that the notation is a bit confusing, and the paper's layout is somewhat disorganized, making it hard to follow. I would suggest that the authors provide a guideline at the end of the introduction to help readers understand the role of each section.

---

> ### Author Rebuttal · Authors · 2023-08-09
>
> ## Response to Reviewer b3yr
>
> - **W1**: I have to say that the notation in this paper is confusing, which imposes a burden on me. For instance, h is defined as a node feature on line 81, and f is a tensor-valued function on line 70. However, f is then used as a bijection on line 112. Does this mean the definition of h on line 112 is different from that on line 81?
>
> Sorry for the heavy notation. The $h$ on line 112 and line 81 denotes the node features. However, the repeated use of $f$ in line 70 and 112 may cause a burden for readers. Do you think it’s better if we use $\varphi$ to denote the bijection map rather than $f$?
>
> - **W2**: Although the analysis of the expressivity of 3D GNNs is a first, this paper seems to be based on the analysis in [1]. Moreover, the method proposed by the analysis in [1] can incorporate structural properties into many GNNs, such as GCN, GIN, and GAT. Can this paper similarly incorporate mutual 3D substructures into other 3D GNNs?
>
> Thank you for your insightful question. Indeed, [1] has laid the groundwork with a novel perspective for crafting potent GNNs on 2D graphs. Building upon this inspiration, we extend our analysis to the realm of 3D graphs. The primary distinction between 2D and 3D graphs lies in the inclusion of not only nodes (with features) and edges (with or without features), but also 3D coordinates for each node. **It's exactly the 3D symmetry that makes our extention nontrivial.**
>
> Regarding various 3D GNNs: Yes, our proposed method can seamlessly integrate 3D geometric information into diverse 3D GNNs, provided that geometric features such as distances, angles, or equivariant steerable features are integrated. Specifically, the message passing block illustrated in Figure 3 (between LSE and FTE) can be applied to any equivariant 3D GNN (e.g., Schnet, GVP-GNN, EGNN, or invariant graph attention messgae passing).  The equivariance of our method is guranteed by theorem 5.1. Furthermore,
> In our experimental evaluation, we employ a previous sota method (PaiNN) as the backbone message passing scheme and enhance its capabilities by incorporating our LSE and FTE blocks.
>
> The adaptability of our approach across different 3D GNN architectures underscores its versatility and potential impact on advancing the field.
>
> - **W3**: Recently, some work [2] has analyzed the expressive power of GNNs from the perspective of the attention mechanism, so how can we analyze the expressive power of 3D GNNs from the attention mechanism?
>
> Thanks for bringing up this nice work.  In fact, LEFTNet also implement the structural coefficients in theorem 3.1 as a set of weights that are multiplied to the node features. Note that attention mechanism can be also seen as defining a set of weight, however the difference is that the attention coefficients go through the softmax normalization. Even though we do not see an immediate connection to transfer in [2] to 3D Equivariant GNNs, we do see it is a promising direction to enhance our understanding of 3D Equivariant GNNs and improve interpretability. We will leave this as future work.
>
> - **W4**: In my personal opinion, one of the limitations of this paper is that the notation is a bit confusing, and the paper's layout is somewhat disorganized, making it hard to follow. I would suggest that the authors provide a guideline at the end of the introduction to help readers understand the role of each section.
>
> Thanks for the great suggestion! We will reorganize the structure with a roadmap at the end of the introduction.
>
> [1] A new perspective on" how graph neural networks go beyond weisfeiler-lehman?"
>
> [2] Interpretable and generalizable graph learning via stochastic attention mechanism.

---

> > ### Comment · Reviewer_b3yr · 2023-08-10
> >
> > Thanks for your response. For the notation, I think $\varphi$ is ok. My concerns have been addressed. And I suggest to improve the writing and reorganize some parts.

---

> > > ### Author Response · Authors · 2023-08-10
> > > **Thanks**
> > >
> > > Thanks again for your suggestion! We will make sure to improve organization in the revised version.

---

### Official Review · Reviewer_gxFS · 2023-07-06

**Soundness:** 3 good
**Presentation:** 2 fair
**Contribution:** 3 good
**Rating:** 7
**Confidence:** 3

**Summary:**

This paper introduces local structure encoding and frame transition encoding for more expressive representation learning in 3D graph neural networks. The local structure encoding is inspired by observations in the proposed local hierarchy of 3D graph isomorphisms; specifically the observation that subgraph isomorphism has greater discriminative power than triangle or tree isomorphism. The frame transition encoding is inspired by the observation that not all types of invariant interactions between disjoint clusters can be expressed as functions of the union of those clusters. The authors show that the proposed 3D GNN exceeds the performance of or performs on par with a multitude of existing models.

**Strengths:**

* Originality: The proposed work appears original. The authors introduce a classification of local and global structure, and use it to classify existing 3D GNNs. They also show how to incorporate the most expressive types of local and global structural elements into a new 3D GNN and show that it outperforms/performs on par with existing methods.
* Quality/Clarity: The proposed work is of good quality. The paper is well written, preliminary ideas are presented in an accessible way; the local hierarchy of 3D graph isomorphisms is paired with a very nice figure; the work is well motivated; the proposed model outperforms/performs on par with existing methods.
* Significance: The problem of building expressive representations in GNNs is challenging and of significance to the community.


**Weaknesses:**

* Quality: see questions

**Questions:**

* SpookyNet is not cited or discussed although it shows very strong performance in the analysis. This method appears to have local and global features, it would be interesting to see where it lands in the proposed classification.

**Limitations:**

The authors discuss some limitations of the method

---

> ### Author Rebuttal · Authors · 2023-08-09
>
> ## Response to Reviewer gxFS
>
> Thanks for your meticulous examination and insightfull feedback.
>
> **Q1**: SpookyNet is not cited or discussed although it shows very strong performance in the analysis. This method appears to have local and global features, it would be interesting to see where it lands in the proposed classification.
>
> **Response**:
> Thank you for bringing this to our attention. We apologize for the oversight. In Table 2, we have compared our method with SpookyNet, but regrettably, we neglected to cite it in the main text. We will correct it in the revised version.
>
> SpookyNet is a strong baseline with both local and nonlocal interaction blocks. But **the meaning of “local” and “nonlocal” in SpookyNet are different from the local-to-global analysis in our paper.**
>
>   - Specifically, we focus on the expressive power of 3D GNNs, in other words, the ability to distinguish different 3D structures. Here local means local 3D structures, and global means the whole input 3D structure, as used in [1].
>
>   - While in SpookyNet, **the local interaction aims to incorporate neighboring information into central nodes**, which is essentially what an MPNN layer can do. The nonlocal interaction aims to capture long-range interactions between nodes which can not be captured by an L-layer GNN, therefore, they **use attention to consider interactions between all pair of nodes**. This local and nonlocal idea is similar to the method in this paper [2].
>
> In short, we focus on the ability to distinguish different local and global structures, while SpookyNet focuses on incorporating local and global information. These two are different.
> For example, a SchNet-like model which considers edge distances during message passing indeed can capture local interactions, but it cannot distinguish different local structures.
>
> It is worth mentioning that instead of simply using a SchNet-like model to capture local interactions, SpookyNet uses a more powerful model, as shown in Figure 3 of their paper. Specifically, it first constructs some basis functions based on Bernstein polynomials and spherical harmonics. These basis functions are then used to update node features in the local interaction block. Putting SpookyNet in Table 4, we can say that it satisfies LSE, but only partially satisfies FTE by using the nonlocal interaction. Note that it doesn't update equivariant features. The final outputs of the local interaction block are only invariant features.
>
>
>
> [1] ComENet: Towards complete and efficient message passing for 3D molecular graphs
>
> [2]. Recipe for a general, powerful, scalable graph transformer

---

> > ### Comment · Reviewer_gxFS · 2023-08-17
> >
> > Thank you for your response. I will keep my score.

---

### Official Review · Reviewer_7xD9 · 2023-07-06

**Soundness:** 3 good
**Presentation:** 3 good
**Contribution:** 3 good
**Rating:** 7
**Confidence:** 4

**Summary:**

Authors analize the expressive power of 3D equivariant GNNs and introduce a new expressive equivariant GNN architecture based on local node and edge wise frames.

**Strengths:**

The proposed way to achieve equivariance is well grounded and conceptually very interesting and novel.
The construction is also based on solid theoretical motivation and achieved good results in practical experiments.
The paper is well written

**Weaknesses:**

The theory about the local hierarchy of 3D Graph Isomorphism (section 3) relatively straightforwardly follows from existing works on equivariant GNN expressive power and subgraph GNN expressive power. That being said, the overall package is still very solid.

**Questions:**

The local frame construction reminded me a bit of Frame Averaging by Puny et al. (ICLR 2022). It would be interesting to see how they stack up.

**Limitations:**

Authors have addressed the limitations

---

> ### Author Rebuttal · Authors · 2023-08-09
>
> ## Response to Reviewer 7xD9
>
> Thanks for your meticulous examination and insightfull feedback.
>
> **Response to the Weakness**:
>
> We appreciate your discerning observation and insightful feedback regarding the theoretical aspect of the local hierarchy of 3D Graph Isomorphism in Section 3. Indeed, the foundation of this theory draws upon existing works on 2D GNN expressive power and subgraph GNN expressive power.
> However, it's important to note that even within the context of established theories, the synthesis and customization of these ideas to our specific problem domain contribute to the overarching value of our work.
>
> We would like to emphasize the coherent theoretical logic that emerges from the synergy between Section 3, Section 4, and the neural sheaf interpretation in appendix. To enhance the clarity of this logical progression, we will incorporate a roadmap at the conclusion of the introduction, highlighting the interconnectedness of these sections.
>
> **Q1: Frame Averaging by Puny et al. (ICLR 2022)**
>
> A1: Thanks. After carefully checking, we believe Puny et al. (ICLR 2022) [1] provided another way of building equivariant frames.  However, their specific frames come out of the PCA decomposition, which are essentially global. Then, the averaging is a weighted sum of these global frames (a finite approximation of integration on the Lie group).
> We will add the discussion of this work into our related work section.
>
> [1] Frame Averaging for Invariant and Equivariant Network Design

---

> > ### Comment · Reviewer_7xD9 · 2023-08-17
> >
> > Thank you for your response. I will keep my score.

---

### Official Review · Reviewer_Wt9q · 2023-07-14

**Soundness:** 3 good
**Presentation:** 2 fair
**Contribution:** 2 fair
**Rating:** 5
**Confidence:** 3

**Summary:**

This paper investigates expressive message passing architectures for processing 3D geometric graphs with permutation and Euclidean symmetries. The authors first provide an investigation of a local hierarchy of isomorphism separability. Then, the authors show cases where powerful local invariant models may fall short of encoding global structures due to information loss during propagation, and aim to reduce the information loss by turning to equivariant message passing incorporating frame transition matrices that are related to neural sheaf diffusion. Based on the theoretical framework, the authors propose a message passing architecture that is expressive in terms of the local isomorphism separability as well as incorporating frame transition. Experimental results on invariant molecular property prediction on QM9 and equivariant force prediction on MD17 shows that the proposed approach outperforms previous equivariant neural networks.

**Strengths:**

S1. This paper addresses an important problem of understanding the theoretical expressiveness of geometric graph neural networks and developing architectures that are provably expressive.

S2. The local isomorphism hierarchy proposed in Section 3 is novel as far as I can tell (in context of 3D geometric graphs) and correct.

S3. The visual example shown in Figure 1 was helpful in understanding the proposed hierarchy of local isomorphism.

S4. The connection of the proposed approach to sheaf diffusion mentioned in Appendix I was interesting.

S5. The experimental results and ablation study on QM9 and MD17 seems to support the main arguments.

**Weaknesses:**

W1. The equality in Eq. 3 seems not correct as it neglects permutation symmetry. If we consider positional features represented as matrices $\mathbf{X}\in\mathbb{R}^{n\times 3}$ (Line 77), the precise symmetry under consideration is a direct product of permutation group and Euclidean group $S_n \times SE(3)$ (Dym et al., 2020). I think it should be clarified that the equality in Eq. 3 is up to permutation of nodes.

W2. In Line 111, I think  {-tree, -triangular, -subgraph} should be {tree-, triangular-, subgraph-}. Same applies to Line 603-604 and Line 612.

W3. In the definition of tree and triangular isometries (Line 114-118), is there any reason to use the term tree and triangular? I think the definitions are a straightforward extension of the subtree and overlap isomorphism proposed in Wijesinghe et al., 2022, respectively, and I see no reason to use different name. Especially, for the triangular isometry, I found the name misleading since the isometry condition is imposed on the entire set of triangles that share $e_{ui}$, rather than individual triangle.

W4. In the definition of triangular isometry (Line 116), the condition is specified for each edge in the intersection of the local subgraphs $e_{iu}\in\mathbf{S}\_{i-j}$. Please correct me if I am wrong, but to make sense, I think the condition should be for each edge in one of the local subgraphs, $e_{iu}\in\mathbf{S}\_{i}$.

W5. In Line 606, the term structure completeness is used without proper definition (although it is emphasized by boldface). Without clear definition, I was not able to properly understand and correctness of the argument in Line 608.

W6. The argument in Line 162-164 is ambiguous; do you mean a composition of Atoministc Cluster Expansion and invariant map constructed using scalarization by edge-wise equivariant frames is a universal approximator of SE(3) invariant functions, so it can serve as $\phi$ in Theorem 3.1? Is the proof of universality available in literature?

W7. For LSE, based on Appendix H, it seems that the hierarchy of local isomorphism proposed in this paper is a subcase of the geometric WL test proposed in Joshi et al., 2023. Since this paper provides 3-way classification while GWL provides a more sensitive separation of expressive power of geometric GNNs based on the orders of node tuple and feature tensors, one can argue that the local isomorphism hierarchy proposed in this paper is less of practical interest compared to GWL.

W8. For FTE, in Section 4, frame transition is introduced to solve the limitation of SE(3) invariant messages as outlined in Theorem 4.1. But as far as I know, a majority of practical equivariant geometric GNNs in literature are already using equivariant (tensor) messages (Thomas et al., 2018; Fuchs et al., 2020; Satorras et al., 2022; Geiger et al., 2022; Kohler et al., 2019), and the expressive power of some of them (TFN (Thomas et al., 2018) and SE(3)-Transformers (Fuchs et al., 2020)) are shown to be universal i.e., maximally expressive (Dym et al., 2020). Is the motivation of frame transition also relevant for these architectures? Also, is there specific reasons to favor the proposed frame based propagation over these already equivariant message passing algorithms, despite the added complexity of incorporating node- and edge-wise frames?

W9. In Line 214, does the frame here refer to frames in Riemannian manifold theory? Since there is no proper definition (Eq. 5 provides a description of key property but is not a definition), I was confused on what exactly are the frames mentioned here and was unable to verify whether Line 216-221 is correct. It seems the precise definition and descriptions of equivariant frame are completely deferred to Appendix B and F although it serves as a main component of the proposed architecture, which makes the paper hard to read and understand; I think key parts of the proposed algorithm should be described in the main text in a self-contained manner.

W10. Overall, I find that the writing of the paper has room for improvement in terms of organization and readability. There are many cross-references back and forth across the paper, so I couldn't serially read through each section without going through a large portion of the entire text multiple text. For example, Line 236-237 (Section 4) refers to Figure 3 (Section 5), Line 555 (Appendix A) refers to Eq. 21 (Appendix F), Line 592 (Appendix C) refers to Figure 3 (Section 5), Algorithm 1 (Appendix C) refers to Eq. 4 (Section 3), and so on.

W11. In Eq. 7, I don't see how the equation describes equivariance. I think it should be something like $\mathbf{m}(g\mathbf{x}\_u) = \sum\_{i=0}\^l\mathcal{M}\^i(g)\mathbf{m}\_i(\mathbf{x}\_u)$, is this a typo?

W12. Typo in Line 1 of Algorithm 1, gragh -> graph

W13. Specification of the output is missing in Algorithm 1, and the output in Algorithm 2 is not clear since normal and boldfaced h are mixed in the algorithm. Furthermore, In Line 7 of Algorithm 1, how do you scalarize the mutual 3D structure, which is a subgraph, based on edge frame $F_{ij}$? I don't see how this can be done in Eq. 8 which defines scalarization. Similarly, in Line 6 of Algorithm 2, how do you scalarize the mutual 3D structure without frame? Also, if the algorithm 2 is LSE only, why should one compute edge-wise frames in Line 5?

W14. In the proof of Theorem 5.1, I can see that the proposed parameterization is universal, but does it correctly guarantee equivariance as well (as specified in Theorem 5.1)? The current proof seems to clearly show universality, but I am not sure about equivariance.

Dym et al., On the universality of rotation equivariant point cloud networks (2020)

Wijesinghe et al., A new perspective on "how graph neural networks go beyond weisfeiler-lehman" (2022)

Joshi et al., On the Expressive Power of Geometric Graph Neural Networks (2023)

Thomas et al., Tensor field networks: Rotation- and translation-equivariant neural networks for 3D point clouds (2018)

Fuchs et al., SE(3)-Transformers: 3D Roto-Translation Equivariant Attention Networks (2020)

Satorras et al., E(n) Equivariant Graph Neural Networks (2022)

Geiger et al., e3nn: Euclidean Neural Networks (2022)

Kohler et al., Equivariant Flows: sampling configurations for multi-body systems with symmetric energies (2019)

**Questions:**

The questions are included in the weakness section.

**Limitations:**

The authors have addressed the limitations of their work in Section 7.

---

> ### Author Rebuttal · Authors · 2023-08-09
>
> ## Response to Reviewer Wt9q
>
> > typos (W2, W4, W11, W12, W13 (Alg.1 and 2))
>
> Thanks for your meticulous examination and useful feedback. We will fix them in the revised version.
>
> > Other questions
>
> **Due to 6k characters' limitation, we only provide short answers. We are happy to explain more if you have further questions. In addition, discussions about frame transition and its realizations (W8 and W9) are provided in the [general response](https://openreview.net/forum?id=hWPNYWkYPN&noteId=1Xmyobc4x8).**
>
> - W1: Eq.3 permutation symmetry: Will clarify that Eq. 3 is up to the permutation of nodes
>
> - W3: name of the isomorphisms
>
> The sub-tree isometry corresponds to our tree isomorphism as the name suggests. However, the overlap isomorphism is subtle, in the sense that overlap is an obscure terminology for formal definitions. This obscurity becomes more severe when both 2D and 3D structures are present. **Therefore, we propose a more intuitive “ triangular” as an alternative.** But, you are right that triangular isomorphism is based on the entire set of triangles that share (a bundle of triangles that share a common edge). We will add a remark to directly point out that the triangular isomorphism corresponds to 3D overlap isomorphism.
>
> - W5: definition of structure completeness
>
> Sorry for the ambiguity. The structure completeness in Appendix refers to the ability to classify non-isomorphic 3D structures (the formal definition is in [1]). In other words, if a neural network can classify non-isomorphic 3D graphs into different categories, we say that this neural network is structure complete. The reason we add the restrictive prefix 'structure' is to emphasize that this completeness doesn't imply a neural network has the universality of approximating any continuous functions as Sec.4 discussed.
>
> - W6: about Line 162-164
>
> 1 and 2 are two parallel ways to achieve local universal invariant approximator: 1. Applying local frames will first transform all equivariant quantities to invariant scalars (this procedure doesn’t lose information, as this transform is invertible, see [2], then the rest of the argument follows the universality of MLPs. 2. Atoministc Cluster Expansion can express equivariant functions universally, then the invariant encoder can be built by adding another projection layer (that transforms equivariant output to invariant scalars). This layer can be found in [3] and since ACE is composed of taking the tensor product of equivariant vectors for all orders, and the tensor product type of universality has also been proved in [3], and therefore we neglect some details.
>
> - W7: compare to GWL
>
> Thanks for asking this insightful question. First of all, our setup is different from [4] but highly related. We seek to implement expressive (and efficient) equivariant graph neural network that avoids higher-order message passing scheme. Thus, we only considered one-hop message passing. The high level philosophy of our local approach is similar with [5], which also goes beyond WL by encoding local structural coefficients. On the other hand, [4]’s theory is inspired by classical potential theory of many body interaction systems, which is complementary to our geometry view.
> Secondly, we are inspired by [5] such that we aim to develop a local hierarchy of isomorphism that could characterize the expressiveness of 3D equivariant GNNs in a finer scale. We actually believe our local isomorphism hierarchy would be of practical interest to the community in the sense that we can quickly design efficient algorithms (e.g., LEFTNet) to improve expressiveness within one-hop region. We do agree a concrete technical connection between the local isomorphisms and GWL would be interesting (such as Theorem 2 in [5] as a 2D analogy) for future works as GWL is also a recent work.
>
> - W13:
>
> Alg.1: We will add the detailed output, which should be the output block(see the framework in fig.3) which takes all node features (invariant & equivariant) for final prediction.
>
> Alg. 2: Typo here. All features should be invariant.
>
> Line 6 of Alg.2: Typo here. It should be the same as line 7 of Alg.1.
>
> The scalarization step (details in Fig.3 and line 258-265):
> The scalarization of a subgraph is realized by scalarizing all the 3D position vectors within this subgraph. In summary, the mutual 3D structure $S_{i-j}$ is associated with edge $e_{i-j}$, and we build an equivariant frame for each edge. Then, we scalarize all the 3D particles inside $S_{i-j}$ through this edge frame by Eq. 8. In summary, LSE requires edge equivariant frames, and FTE requires node frames in our implementation.
>
> W14: equivariance of Theorem 5.1
>
> Since the key ingredient of proving equivariance is similar with previous [2]'s appendix (also by chasing the lower part of the commutative diagram in Line 786), we neglect some details on the equivariance side of the proof. Now we are happy to fill in the details. Roughly speaking, the equivariance is proved by showing that the composition of scalarization, the MLP, and the tensorization is equivariant. The equivariance of the composition of scalarization and MLP is obvious since scalarization turns equivariance into invariance, and then the MLP is a transformation of invariant scalars. On the other hand, the tensorization defined in Line 779-782 is a pairing of equivariant tensors and invariant scalars, which is equivariant by definition. Combining the two steps, we have proved that our implementation of FTE is equivariant.
>
> **Ref**
>
> [1] ComENet: Towards complete and efficient message passing for 3D molecular graphs.
>
> [2] SE (3) equivariant graph neural networks with complete local frames
>
> [3] On the Universality of Rotation Equivariant Point Cloud Networks
>
> [4] On the expressive power of geometric graph neural networks
>
> [5] A new perspective on" how graph neural networks go beyond weisfeiler-lehman?"
>
> **Note that we provide the response of W8-W10 in the general response.**

---

> > ### Author Response · Authors · 2023-08-15
> >
> > Dear Reviewer Wt9q,
> >
> > Thanks again for your additional questions. We are eager to hear your thoughts about our response! We are also happy to discuss further if you still have any concerns.
> >
> > Sincerely,
> >
> > Authors

---

> > > ### Comment · Reviewer_Wt9q · 2023-08-16
> > > **Response to the comment**
> > >
> > > Hello, thank you for the reminder. My major concerns were (in summary) (1) readability and clarity of the manuscript, and (2) the contributions of the proposed framework compared to GWL and existing SE(3)/E(3) equivariant message passing. With the response, I feel that the readability and clarity of the manuscript will be improved, while it might require a non-trivial amount of revisions. For comparison with message passing, the additional reports on practical optimization benefits seems interesting and necessary (since existing tensor message passing is theoretically already universal), and I highly recommend to organize and include such results in the manuscript. I am currently inclining towards acceptance of the paper, except that I expect the above revision to change the manuscript significantly and I am not able to review the revised version in this round. I have updated my score accordingly.

---

> > > > ### Author Response · Authors · 2023-08-16
> > > >
> > > > We will make the suggested revisions as promised in our updated manuscript. We sincerely appreciate your comments to help improve the clarity and quality of our paper!

---

### Author Rebuttal · Authors · 2023-08-09

## General Response

We thank all the reviewers for their valuable comments and appreciate that all the reviewers find our study novel and interesting for 3D Equivariant GNNs. As mentioned by the reviewers,
- the problem we are focusing on is challenging and of significance to the community (Reviewer Wt9q and gxFS),
- our work is novel and original (all reviewers),
- we have solid theoretical analysis (Reviewer 7xD9),
- our paper is well-written with nice figures (Reviewer Wt9q, 7xD9, and gxFS),
- our method achieves good results, and the ablation study can support the main arguments (all reviewers).

We also would like to thank Reviewer Wt9q and Reviewer b3yr for pointing out some typos and repeated notation that may cause misunderstandings for general ML audiences. We will fix them in the revised version.

In addition, we will add a roadmap at the end of the introduction, and rearrange the order of the appendix to avoid some cross-references, following reviewer Wt9q and b3yr’s suggestion.

Here we add additional explanation about frame transition (FT) (**suggested by Reviewer Wt9q**),  since FT is also one of our main contributions.

W8: about frame transition: why do we need FT?

Our introduction of frame transition (FT) addresses a universal local-to-global phenomenon intrinsic to equivariant neural networks. In this sense, FT is a concept  designed for almost **all equivariant graph neural networks that implement local message passing as backbone models**. On the other hand, this concept assumes significance in the context of symmetry-aware processing, and there won’t be any frame (or coordinate) ‘transitions’  in non-equivariant neural networks used for 3D point clouds, which operate without considering symmetry.

We acknowledge your observation regarding the prevalent use of equivariant architectures that implicitly incorporate aspects of frame transition. In fact, Table 4 of our work elucidates how these architectures partly encode frame transition information, thus **offering a principled basis for their empirical success**. Two of the relevant equivariant models you mentioned have inadvertently been omitted. We will promptly rectify this oversight by including them in Table 4.

To complete our logical framework, we emphasize that invariant architectures can fully capture frame transition information as well. You will find a computationally efficient realization of this principle in Section H of the supplementary materials, where we explore the incorporation of neural sheaf and combined node-edge equivariant frames for a pure invariant encoding of FT.

Regarding the **preference for our proposed frame-based propagation over existing equivariant message passing algorithms**, we underscore three key practical advantages: **1**. Our approach explicitly encodes Frame Transition (FT), offering **both equivariant and invariant methods** for FT encoding, as mentioned earlier. Furthermore, the expressiveness of our FT-encoding is guaranteed by theorem 5.1;
**2**. Optimization Benefits: Our empirical findings demonstrate enhanced stability and faster training for our LSE + FTE modules. While expressiveness gauges approximation errors, optimization challenges persist in neural networks [2]. Hence, we advocate exploring diverse FT realizations to navigate optimization complexities;
**3**. Multi-Modality Pretraining Ease: We observe that incorporating invariant architectures with other neural networks is seamless due to invariant scalar hidden layers. These layers can seamlessly integrate with invariant representations from other modalities [1]. Consequently, the invariant encoding of FT could confer advantages in such multi-modal pretraining scenarios.

[1] Learn to Combine Modalities in Multimodal Deep Learning

[2] DeepONet: Learning nonlinear operators for identifying differential equations based on the universal approximation theorem of operators

W9: about frame transition: definition of frames

We will move the definition of equivariant frames to the main text. In line 214, the orthonormal frames and especially the orthonormal property are defined in terms of the ordinary Euclidean metric. Therefore, no Riemannian metric is needed. We only mention Riemannian geometry and the frame bundle in the limitations and future work section, as our local frame has a root in the Riemannian context as a basis of the tangent space attached to spatial points.

W10: paper organization

Apologies for the recurrent cross-references throughout the paper. We have taken steps to enhance this aspect. Given that our paper encompasses both theoretical groundwork and theory-inspired algorithm design, and the theory itself encompasses two intricately linked components—1. Local Analysis and 2. Local-to-Global Intermediate Analysis—cross-references become unavoidable and imperative to ensure the rigor of the article.

To mitigate potential confusion, our approach involves furnishing intuitive explanations for each mentioned concept while reserving detailed arguments for the appendix (please consult the supplementary materials' version of the appendix, which is further refined). As part of our improvement strategy, we intend to: 1. Enhance the paper's organization by providing a more structured "roadmap" at the conclusion of the introduction section.
2. Relocate the **initial portion of Appendix F (as featured in the supplementary materials) to Line 556** to eliminate the need for a cross-reference.

---

> ### Comment · Reviewer_Wt9q · 2023-08-11
> **Some additional questions**
>
> I appreciate the response from the authors. I want to ask for some additional clarifications:
>
> Regarding optimization benefits in the response to W8, can you elaborate a bit more? In which ways did you observe enhanced stability and faster training for LSE + FTE compared to existing equivariant message passing algorithms? I might have missed it, but cannot find relevant results in the main text or supplementary material.
>
> Regarding ease of multi-modality pretraining in the response to W8, can you provide some example-based description? I looked up [1], but couldn't find evidences on how invariant scalar hidden layers can be easier to integrate across modalities.

---

> > ### Author Response · Authors · 2023-08-11
> > **Further response**
> >
> > Thanks for the quick response.  We would like to make more clarifications on your further concerns.
> >
> > **optimization benefits**: Thanks for posing this question. Here, we empirically find that our LEFTNet is not sensitive with respect to Lr scheduler.  We have tried ReduceLROnPlateau, StepLr, warm-up, and LEFTNet achieved quite similar results. While some baseline methods are relatively sensitive with respect to optimization tricks. Note that this empirical finding is informal and may also depend on datasets, and we cannot test all optimization tricks for all baseline methods, so we didn't include it in the paper.  In fact,  different algorithms may reveal different optimization benefits for different datasets. Therefore,  we advocate exploring diverse FT realizations（including our realizations) in Table 4 to navigate optimization complexities. In conclusion, we will emphasize in section 7 that our theoretical analysis in this paper is restricted to the expressiveness power, while the optimization error for different algorithms remains a challenge that is worth future investigation.
> >
> > **integrate across modalities**: In the earlier response, we provided a general reference on multi-modality training. Here, we provide a multi-modality pre-training specific to 3D structures: [2]. In [2], the authors provide a general pretraining framework by maximalizing the mutual information between the 2D modality and 3D  modality of molecules.  Note that the contrastive learning between the latent representation of 2D and 3D in [2] must be invariant, otherwise, the EBM-NCE contrastive loss (the first part of Eq. 13 of [2]) itself no longer respects the symmetry.  Since the 2D representation (usually comes from an ordinary GNN)  is automatically invariant, the 3D latent representation paired with the 2D representation should also be invariant.
> >
> >
> > [2] A Group Symmetric Stochastic Differential Equation Model for Molecule Multi-modal Pretraining,   ICML 2023

---

### Decision · Program_Chairs · 2023-09-21

**Decision:**

Accept (poster)

**Comment:**

This paper suggests a novel approach for building 3D equivariant graph neural networks. First, it develops hierarchy of local 1-hop subgraph isomorphisms to test local expressiveness; and second, it discusses how to aggregate local information globally with minimal loss of information leading to the incorporation of frame transition matrices. On the downside, the main theoretical contributions of this paper closely follow previous works, and the benefit over alternative 3D equivariant architectures does not seem very large.